# Dynamic Inference with Neural Interpreters

**Muhammad Waleed Gondal**[*,1]   **Nasim Rahaman**[*,1,2,3,‡]   **Shruti Joshi**[1]   **Peter Gehler**[3]

**Yoshua Bengio**[†,2]   **Francesco Locatello**[†,3]   **Bernhard Schölkopf**[†,1]

## Abstract

Modern neural network architectures can leverage large amounts of data to generalize well within the training distribution. However, they are less capable of systematic generalization to data drawn from unseen but related distributions, a feat that is hypothesized to require compositional reasoning and reuse of knowledge. In this work, we present Neural Interpreters, an architecture that factorizes inference in a self-attention network as a system of modules, which we call *functions*. Inputs to the model are routed through a sequence of functions in a way that is end-to-end learned. The proposed architecture can flexibly compose computation along width and depth, and lends itself well to capacity extension after training. To demonstrate the versatility of Neural Interpreters, we evaluate it in two distinct settings: image classification and visual abstract reasoning on Raven Progressive Matrices. In the former, we show that Neural Interpreters perform on par with the vision transformer using fewer parameters, while being transferrable to a new task in a sample efficient manner. In the latter, we find that Neural Interpreters are competitive with respect to the state-of-the-art in terms of systematic generalization.

## 1   Introduction

Rule-based programming is the basis of computer science, and builds the foundation of symbolic-AI based expert systems that attempt to emulate human decision-making to solve real-world problems. The process of inference entails channeling information through a chain of computational units (e.g., logical primitives) and culminates in a conclusion that answers a given query. Such systems have the advantage that they permit efficient reuse of computational units and enable iterative reasoning over multiple cycles of computation. As a simple example, consider the relation `parent_child(A, B)`, which can be used to construct a new relation `sibling(U, V) = parent_child(A, U) AND parent_child(A, V)`, which in turn can be used to construct yet another relation `cousin(U, V) = parent_child(X, U) AND parent_child(Y, V) AND sibling(X, Y)`, and so on. However, such systems are known to suffer from the knowledge acquisition problem, which is the inability to leverage unstructured data to derive new computational units and improve existing ones [30].

In stark contrast to the symbolic paradigm, modern machine learning models excel at absorbing large amounts of unstructured data and yield strong performance in many challenging domains, ranging from large-scale image classification to language modelling. However, they are relatively rigid in how they share and reuse computation in order to process information: convolutional neural networks, for instance, process the content of an image at every location to arrive at the class label of a given image. In doing so, they only reuse computational units (here, convolutional filters) *laterally* at a constant depth i.e., amongst information being processed simultaneously. In the same spirit, recurrent neural networks only reuse computational units (here, RNN cells) *vertically*, i.e., in depth. Such rigidity

---

[*,†]Equal contribution, ordered alphabetically by last name. [1]Max-Planck-Institute for Intelligent Systems, Tübingen. [2]Mila, Québec. [3]Amazon Web Services. [‡]Work partially done during an internship at Amazon Web Services.

35th Conference on Neural Information Processing Systems (NeurIPS 2021).

in how computation is reused is believed to be one of the reasons current deep neural networks are less capable of systematically generalizing to problems not encountered during training [4, 33, 39].

In the present work, we draw inspiration from typed programming languages to develop a self-attention based neural architecture that relaxes this rigidity in computation reuse. The resulting class of models, which we call Neural Interpreters, *learns* to flexibly use and compose computational units directly from data without additional supervision. Neural Interpreters factorize a self-attention network [58] as a system of computational units that we call *functions*. The input to the network is set-valued and processed in the network by a dynamically inferred sequence of functions – potentially by the same function multiple times, enabling vertical sharing of computational units. This aligns with earlier ideas on independent mechanisms [21–24, 42, 43, 46, 51]: a set of mechanisms can be combined and reused in many different ways depending on context (the current input or task), thus factorizing knowledge in independent pieces which can lead to better systematic generalization.

Neural Interpreters have two key benefits over vanilla self-attention networks. First, the modular inductive biases facilitate generalization beyond the training distribution and adaptation to new tasks in a sample-efficient manner. Second, but consistent with the notion of factorizing knowledge into approximately independent and composable pieces, the proposed parameterization is (by construction) not only agnostic to the cardinality of the input set, but also to the number of functions. The latter implies that given a new task, additional functions can be non-disruptively added and fine-tuned. In other words, knowledge acquired from the prior training tasks can be effectively repurposed for new tasks.

**Our primary contributions are as follows. (a)** We introduce the Neural Interpreter, an attention-based architecture that can be applied on arbitrary set-valued inputs or representations. **(b)** We quantitatively evaluate the proposed architecture in two distinct problem settings: multi-task image classification and abstract reasoning. In the former setting, we show that Neural Interpreters are capable of sample-efficient adaptation to new tasks and can exploit additional capacity added after pre-training. In the latter setting, we demonstrate that Neural Interpreters are capable of out-of-distribution generalization. In particular, we find that Neural Interpreters outperform the Vision Transformer baseline [13, 17], which in turn is competitive in-distribution with the prior state-of-the-art. In both settings, we find that Neural Interpreters develop the ability to gracefully trade-off performance with compute at inference time [63]. In addition, we include results on a toy problem, where we explicitly probe the ability of Neural Interpreters to learn recomposable computational primitives. **(c)** We ablate over the architectural parameters of Neural Interpreters and qualitatively visualize what the model learns. We find patterns in how the input is routed through the network and that a wide range of hyperparameters yield promising results.

## 2   Neural Interpreters

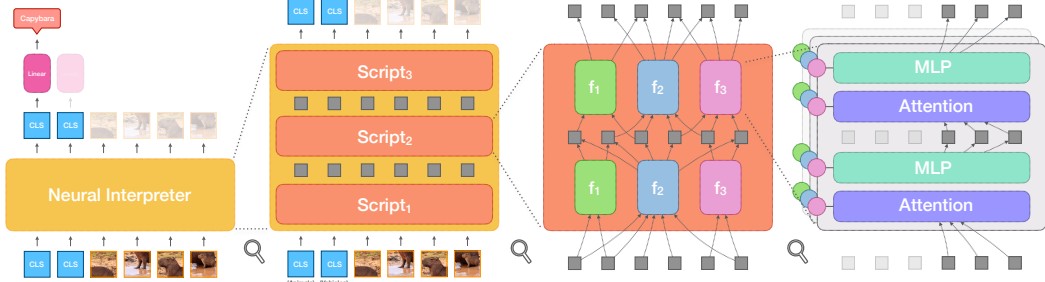

Figure 1: **Leftmost:** Overview of the architecture, shown here with image patches as inputs and two CLS tokens with corresponding classification heads. **Center Left:** The Neural Interpreter, shown here as a stack of three *scripts*. **Center Right:** A script, shown here as a collection of three *functions* applied over two *function iterations*. **Rightmost:** A stack of two *Lines of Code* (LOCs), spread over three parallel computational streams (one per function) and conditioned by the respective function codes (colored circles). Residual connections are present but not shown in this figure.

In this section, we describe in detail the components of a Neural Interpreter; see Figure 1 for an overview. This architecture can be used as a drop-in replacement for a self-attention network, e.g., Transformers [58]. In line with prior work [13, 17, 55], we focus on applications to visual data.

**Input and Output.** The input to the Neural Interpreter is any set with vector-valued elements $\{\mathbf{x}_i\}_i, \mathbf{x}_i \in \mathbb{R}^{d_{\text{in}}}$ and the output is another set of vectors $\{\mathbf{y}_j\}_j, \mathbf{y}_j \in \mathbb{R}^{d_{\text{out}}}$ with the same cardinality as the input set. We assume in this work that the input set contains vector embeddings of image patches [13] or of entire images [5]. The input set additionally includes one or more learned vectors, called CLS Tokens [16], for which the corresponding outputs interface with their respective classifiers [38].

**Scripts.** A Neural Interpreter can be expressed as a stack of $n_s$ *scripts*, where a script maps one set of vectors to another with the same number of elements:

$$\{\mathbf{y}_j\}_j = \text{NeuralInterpreter}(\{\mathbf{x}_i\}_i) = \left[\text{Script}_{n_s} \circ ... \circ (n_s \text{ times}) \circ ... \circ \text{Script}_1\right](\{\mathbf{x}_i\}_i) \quad (1)$$

A script has four components: a type inference module, a type matching mechanism, a set of functions and an interpreter, as explained below. The parameters of these components are not shared between scripts. *Role:* Scripts function as independent building blocks that can be dropped in any set-to-set architecture, and Neural Interpreters with a single script can perform well in practice.

**Functions.** *Functions* make the computational units in Neural Interpreters; they can be represented as a tuple of vector valued parameters, i.e., $f_u = (\mathbf{s}_u, \mathbf{c}_u)$ where $u$ indexes functions. Here, $\mathbf{s}_u$ is referred to as the signature of the function $f_u$ (with a meaning similar to that in programming languages), and it is a normalized vector of $d_{\text{type}}$ dimensions. The signature vector specifies to the type matching mechanism (see below) what inputs are to be routed to $f_u$. We refer to $\mathbf{c}_u$ as the code vector of $f_u$, as it instructs an interpreter module (shared between functions, see below) how to process the inputs to $f_u$. *Role:* Functions are vector-valued *instructions* to other components in the script. They implement the computational units that can be reused in the computational graph.

**Type Matching and Inference.** The type matching mechanism (Figure 2) enables the learned routing of set elements through functions, and it proceeds in three steps. First, given a set element $\mathbf{x}_i$, it is processed by an MLP module that outputs its *type vector* $\mathbf{t}_i$. This module is called the *type inference* module, and the resulting type vector is an element of the same topological space as function signatures, i.e., a $d_{\text{type}}$-dimensional hypersphere $\mathcal{T}$. Next, given a function $f_u$ and its signature vector $\mathbf{s}_u \in \mathcal{T}$, the compatibility $C_{ui}$ between the function $f_u$ and a set element $\mathbf{x}_i$ is determined by the cosine similarity between $\mathbf{s}_u$ and $\mathbf{t}_i$ in $\mathcal{T}$ (larger $\implies$ more compatible). Finally, if this compatibility is

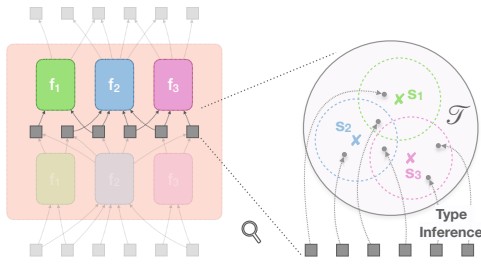

Figure 2: Illustration of the type matching mechanism. Functions only access set elements whose types lie in the vicinity of their signatures.

larger than a threshold ($\tau$), $f_u$ is permitted access to $\mathbf{x}_i$. Formally, let $\{\mathbf{x}_i\}_i$ be a set of intermediate representation vectors (indexed by $i$). With a learnable parameter $\sigma$ and a hyper-parameter $\tau$, we have:

$$\mathbf{t}_i = \text{TypeInference}(\mathbf{x}_i) \in \mathcal{T}; \qquad d_{\mathcal{T}}(\mathbf{s}_u, \mathbf{t}_i) = (1 - \mathbf{s}_u \cdot \mathbf{t}_i) \quad (2)$$

$$C_{ui} = \frac{\tilde{C}_{ui}}{\epsilon + \sum_u \tilde{C}_{ui}} \quad \text{where} \quad \tilde{C}_{ui} = \exp\left[-\frac{d_{\mathcal{T}}(\mathbf{s}_u, \mathbf{t}_i)}{\sigma}\right] \text{ if } d_{\mathcal{T}}(\mathbf{s}_u, \mathbf{t}_i) > \tau, \text{ else } 0. \quad (3)$$

$\tau$ is called the truncation parameter of the kernel and $\epsilon$ is a small scalar for numerical stability. The compatibility matrix $C_{ui} \in [0, 1]$ will serve as a modulation mask [46] for the self-attention mechanism in the interpreter (see below). *Role:* The type matching mechanism is responsible for routing information through functions. The truncation parameter controls the amount of sparsity in routing.

**ModLin Layers and ModMLP.** The components described below make use of linear layers conditioned by some code $\mathbf{c}$. Consider a linear layer with weight matrix $\mathbf{W} \in \mathbb{R}^{d_{\text{out}}} \times \mathbb{R}^{d_{\text{in}}}$ and a bias vector $\mathbf{b} \in \mathbb{R}^{d_{\text{out}}}$. Let $\mathbf{x} \in \mathbb{R}^{d_{\text{in}}}$ denote the input vector to the layer, and $\mathbf{c} \in \mathbb{R}^{d_{\text{cond}}}$ a condition vector, and $\mathbf{W}_c \in \mathbb{R}^{d_{\text{in}}} \times \mathbb{R}^{d_{\text{cond}}}$ a learnable matrix. The output $\mathbf{y} \in \mathbb{R}^{d_{\text{out}}}$ is given as:

$$\mathbf{y} = \text{ModLin}(\mathbf{x}; \mathbf{c}) = \mathbf{W}(\mathbf{x} \otimes \text{LayerNorm}(W_c \mathbf{c})) + \mathbf{b} \quad (4)$$

where $\otimes$ denotes element-wise product and we call the resulting layer a modulated linear layer, or a ModLin layer [3]. Further, one may stack (say) $L$ such ModLin layers (sharing the same condition or code vector $\mathbf{c}$) interspersed with an activation function (we use GELUs [26]) to obtain a ModMLP:

$$\mathbf{y} = \text{ModMLP}(\mathbf{x}; \mathbf{c}) = (\text{ModLin}_L(\,\cdot\,; \mathbf{c}) \circ \text{Activation} \circ ... \circ \text{ModLin}_1(\,\cdot\,; \mathbf{c}))(\mathbf{x}) \quad (5)$$

*Role:* ModLin layers and the ModMLP can be interpreted as *programmable* neural modules, where the *program* is specified by the condition or code vector $\mathbf{c}$.

**ModAttn.** ModAttn is a conditional variant of the kernel modulated dot product attention (KMDPA) [46], where the key, query and value vectors are obtained from ModLin layers conditioned by a vector. In our case, this vector is the code vector $\mathbf{c}_u$ of function $f_u$, and the corresponding key, query and value vectors are computed as follows (with $\{\mathbf{x}_i\}_i$ as input and $h$ indexing attention heads):

$$\mathbf{k}_{uhi} = \text{ModLin}^h_{\text{key}}(\mathbf{x}_i, \mathbf{c}_u) \qquad \mathbf{q}_{uhi} = \text{ModLin}^h_{\text{query}}(\mathbf{x}_i, \mathbf{c}_u) \qquad \mathbf{v}_{uhi} = \text{ModLin}^h_{\text{value}}(\mathbf{x}_i, \mathbf{c}_u) \quad (6)$$

Note that further below (e.g., in Equation 9), we will encounter $\mathbf{x}_{ui}$, where the extra $u$ in the subscript denotes that the set element at index $i$ is specific to the function $u$; in this case, $\mathbf{x}_i$ is substituted with $\mathbf{x}_{ui}$ in Equation 6. Next, given the keys, queries and the function-variable compatibility matrix $C_{ui}$, the modulated self-attention weights $W_{uhij}$ are given by:

$$W_{uhij} = \frac{\tilde{W}_{uhij}}{\epsilon + \tilde{W}_{uhij}} \quad \text{where} \quad \tilde{W}_{uhij} = C_{ui}C_{uj}\left[\text{softmax}_j\left(\frac{\mathbf{q}_{uhi} \cdot \mathbf{k}_{uhj}}{\sqrt{d_{\text{key}}}}\right)\right] \qquad (7)$$

Here, the quantity $W_{uhij}$ denotes the attention weights in function $f_u$ between elements $\mathbf{x}_i$ and $\mathbf{x}_j$ at head $h$ and the softmax operation normalizes along $j$; intuitively, information about $\mathbf{x}_i$ and $\mathbf{x}_j$ is mixed by $f_u$ at head $h$ only if $W_{uhij} \neq 0$. Now, on the one hand, this can be the case only if both $C_{ui}$ and $C_{uj}$ are non-zero, i.e., $f_u$ is granted access to both variables $\mathbf{x}_i$ and $\mathbf{x}_j$ by the typing mechanism. But on the other hand, $f_u$ does not necessarily mix $\mathbf{x}_i$ and $\mathbf{x}_j$ even if both $C_{ui}$ and $C_{uj}$ are non-zero, for the self-attention weights (square brackets) may still be close to zero depending on the context (i.e., the content of $\mathbf{x}_i$ and $\mathbf{x}_j$). Next, the values are linearly mixed using the computed attention weights, which is then processed by a final ModLin layer to yield the output $\mathbf{y}_{ui}$:

$$\mathbf{y}_{ui} = \text{ModLin}(\tilde{\mathbf{y}}_{ui;h}; \mathbf{c}_u) \quad \text{where} \quad \tilde{\mathbf{y}}_{uhi} = \sum_j W_{uhij}\mathbf{v}_{uhj} \qquad (8)$$

Here, $\tilde{\mathbf{y}}_{ui;h}$ means the head-axis is folded into channels. *Role:* ModAttn enables interaction between the elements of its input set in multiple parallel streams, one for each function. The query, key, value, and output projectors of each stream are conditioned on the corresponding code vectors, and the interaction between elements in each stream is weighted by their compatibility with the said function.

**Line of Code (LOC).** An LOC layer is a ModAttn layer followed by a ModMLP layer (Figure 1, rightmost), where both layers share the same condition vector $\mathbf{c}_u$, and there are weighted residual connections between the layers. Assuming inputs $\{\mathbf{x}_{ui}\}_{u,i}$ to the LOC, we have:

$$\tilde{\mathbf{a}}_{ui} = \text{ModAttn}(\{\text{LayerNorm}(\mathbf{x}_{uj})\}_j; \mathbf{c}_u, \{C_{uj}\}_j) \qquad \mathbf{a}_{ui} = \mathbf{x}_{ui} + C_{ui}\tilde{\mathbf{a}}_{ui} \qquad (9)$$

$$\tilde{\mathbf{b}}_{ui} = \text{ModMLP}(\text{LayerNorm}(\mathbf{a}_{ui}); \mathbf{c}_u) \qquad \mathbf{y}_{ui} = \mathbf{a}_{ui} + C_{ui}\tilde{\mathbf{b}}_{ui} \qquad (10)$$

This parameterization ensures that $\mathbf{y}_{ui} = \mathbf{x}_{ui}$ if $C_{ui} = 0$. In words, if a function ($f_u$) is not granted access to a variable ($\mathbf{x}_i$) by the typing mechanism, it acts as an identity function for this variable. Further, note that we allow the input set to be indexed only by $i$; in this case, we assume $\mathbf{x}_{ui} = \mathbf{x}_i$ for all $u$. *Role:* A LOC can be thought of as multiple instances of a layer in the original transformer architecture (comprising a self-attention and a MLP module with residual connections), applied in parallel streams, one per function. Computations therein are conditioned on the respective code and signature vectors.

**Interpreter.** The interpreter layer is a stack of $n_l$ LOCs sharing the same function codes $\mathbf{c}_u$ and function-variable compatibilities $C_{ui}$. Assuming the input to the interpreter is a set $\{\mathbf{x}_i\}_i$, we have:

$$\mathbf{y}_i = \mathbf{x}_i + \sum_u C_{ui}\left(\text{LOC}_{n_l} \circ ...(n_l \text{ times})... \circ \text{LOC}_1\right)(\{\mathbf{x}_j\}_j; \mathbf{c}_u, \{C_{uj}\}_j) \qquad (11)$$

In words: the interpreter broadcasts a given set element to multiple parallel computational streams, one for each function. After the streams have processed their copy of the input element, the results are aggregated by a weighted sum over the streams, where the weights correspond to the compatibility of the input element with the respective function. Equation 11 can be justified by making two observations. First, if a variable $\mathbf{x}_i$ is not matched with any function by the typing mechanism, it is left unmodified by the interpreter; i.e., if $C_{ui} = 0$ for all $u$, then $\mathbf{y}_i = \mathbf{x}_i$. This allows signals to be propagated through the interpreter without interference from existing functions, if so determined by the type inference module. Second, the additive aggregation over the function index $u$ implies that the overall parameterization of Neural Interpreters does not depend on the number of functions. This allows one to add a new function $f_v$ simply by including its signature and code $(\mathbf{s}_v, \mathbf{c}_v)$ as

learnable parameters and finetuning these on (say) a new problem. _Role:_ The interpreter serves as a general-purpose instruction executor (one that is shared between functions). Given a set of inputs and an instruction (here, the function code), it executes said instruction to compute the output.

**Function Iterations in Script.** Having expressed the overall model as a stack of multiple scripts, we are now equipped to describe the computational graph of a single script. A script can be expressed as a recurrent application of the type matching mechanism and the interpreter, where we refer to the composition of the latter two as a Function Iteration (FnIter):

$$\{\mathbf{y}_i\}_i = \text{FnIter}(\{\mathbf{x}_j\}_j) = \text{Interpreter}(\{\mathbf{x}_j\}_j, \{\mathbf{c}_u\}_u, \{C_{uj}\}_{u,j}) \tag{12}$$

$$\text{where } C_{uj} = \text{TypeMatching}(\mathbf{s}_u, \mathbf{x}_j) \tag{13}$$

Here, the TypeMatching component encapsulates both type inference and kernel matching, as detailed in Equation 2. A script (cf. Equation 1) can now be expressed as a recurrent application of FnIter:

$$\{\mathbf{y}_j\}_j = (\text{FnIter} \circ ... \circ (n_i \text{ times}) \circ ... \circ \text{FnIter})(\{\mathbf{x}_i\}_i) \tag{14}$$

_Role:_ Inside a script, function iterations enable sharing of computational units in depth. Increasing the number of function iterations can increase depth without increasing the number of parameters.

**Preventing function signatures and variable types from collapsing on each other.** One might obtain a scenario where the signatures of all functions and the types of all possible set elements collapse to a single point in type-space. This causes all set elements to be routed to all functions, thereby undermining the inductive bias of modularity. One effective way of preventing this degeneracy is to keep the function signatures fixed at initialization (i.e. a high-entropy distribution). This effectively encourages the (learnable) type-inference module to produce diverse types, in order to use all available functions.

**In summary,** we observe that any input element $\mathbf{x}_i$ may trace a large number of _computational paths_ as it progresses through the model, where a computational path is a partially ordered set (poset) of functions (refer to Figure 3 for a visualization of such computational paths). In particular, owing to the fact that we recurrently apply function iterations in scripts, this poset may contain repeated functions, enabling weight sharing in depth. Further, the use of an interpreter that is shared between functions but explicitly conditioned (or _programmed_) by their code vector allows us to remain independent of the number of functions and retain the ability to add more functions after the model has been trained. Future work may investigate manipulating the code vectors themselves, thereby enabling higher-order functions (i.e., functions that manipulate other functions).

## 3 Related Work

**Modularity.** Despite recent successes of deep neural networks, their ability to recombine and reuse meaningful units for systematic compositional generalization is shown to be limited [31, 33, 39]. To mitigate this issue, work has been done to modularize deep architectures to enable the reuse and recombination of learned modules [2, 9, 19, 27, 32, 49, 53], and connections have been drawn to the principle of Independent Causal Mechanisms [7, 22–24, 34, 42, 43, 46, 52]. Modular architectures vary in how the modules are learned and laid out during inference. For instance, Neural Module Networks [2] and Neural Event Semantics [8] dynamically compose the modules using a natural language parser, whereas neuro-symbolic architectures [21–24, 37, 40, 57, 61, 62] typically require entity-centric representations [37, 57, 62] but use neural components like attention for compositional reasoning. Unlike some of these methods, Neural Interpreters (NI) neither require domain knowledge nor object-centric representations. To obtain a strong compositional inductive bias, NIs aim at factorizing the computational graph in terms of functional units which are dynamically recombined in order to solve a particular task. In comparison with SCOFF [23] and NPS [22] which also separate functions, arguments and values and enable a similar dynamic and composed application of functions, NIs introduce the notion of interpreter (which enables parameter sharing between functions and extending the set of functions easily) and sophisticated signature and typing mechanisms described in the previous section.

**Dynamic Routing of Information.** The systematic generalization of modular architectures is shown to depend on the layout of modules [4, 48]. There exist different frameworks to facilitate the joint learning of module layout and parameters [8, 19, 24, 46, 49, 50, 53]. For example, [49] uses collaborative multi-agent reinforcement learned routing, [53] uses a trainable gating network. In this work, we use kernel modulated dot product attention [46] between function signatures and the input types to assign inputs to functions at each step.

**Transformers.** Modularity in Transformers [58] is relatively less studied. Repulsive attention [1] is based on repelling transformer heads to encourage specialization, whereas TIM [34] draws on the principle of independent mechanisms by dividing computation over parallel transformers that interact via competitive top-k attention [24]. In contrast, functions in NIs iteratively acquire set elements with kernel dot-product attention, which, if need be, can uniformly activate all the functions for a given token, encouraging efficient use of computation. Closer to Neural Interpreters, Switch Transformers (STs) [18] employ a mixture of experts framework where experts specialize across tokens, guided by a routing scheme which activates only one expert per token. However, STs focus more on dynamically *selecting* experts, whereas Neural Interpreters focus on dynamically *composing* available experts. The starting point of this work is the vision transformer (ViT) [13, 17], which applies a standard Transformer directly to image patches. We use the same architectural scaffolding as ViTs, but implement additional inductive biases that facilitate fast adaptation and systematic generalization.

## 4 Experiments

In this section, we empirically evaluate Neural Interpreters in a variety of problem settings. The questions we seek to answer are the following. **(a)** Can Neural Interpreters learn reusable computational units, given a number of training tasks that can in principle be solved with a fixed set of primitives? (Section 4.1) **(b)** Do Neural Interpreters modularize information in a way that helps fast adaptation [6]? (Section 4.2) **(c)** Can Neural Interpreters gracefully handle a larger number of functions (computational units) than they were trained with? (Section 4.2) **(d)** Do the inductive biases help with systematic generalization required in abstract reasoning problems? (Section 4.3).

### 4.1 Learning Fuzzy Boolean Expressions

In this section, we construct a toy problem to investigate whether neural interpreters can learn reusable functions when trained on a large number of tasks that share the same underlying building blocks.

**Task Definition.** Consider a set of $N$ scalars $\{x_1, ..., x_N\}$, where $x_i \in [0, 1]$ is a real number between 0 and 1 (inclusive). We now define the following operations on the elements of this set:

$$\texttt{and}(x_i, x_j) = x_i x_j; \quad \texttt{not}(x_i) = \bar{x}_i = 1 - x_i; \quad \texttt{or}(x_i, x_j) = x_i \oplus x_j = \overline{\bar{x}_i \bar{x}_j} \qquad (15)$$

Note that the above operations map from $[0, 1]^2$ to $[0, 1]$; if $x_i, x_j \in \{0, 1\}$, they reduce to their Boolean namesakes. By combining these primitives, it is possible to construct and sample from a family of $2^{2^N}$ *fuzzy* Boolean functions mapping from $[0, 1]^N$ to $[0, 1]$ (Appendix B). The problem is now set-up as follows. We sample 30 random fuzzy Boolean functions of $N = 5$ variables, $\{f_i\}_{i=1}^{30}$ where $f_i : [0, 1]^5 \to [0, 1]$, of which we use 20 for the pre-training phase and reserve 10 for the later adaptation phase. For each function, we sample 163840 points from a uniform distribution over $[0, 1]^5$, of which we use 80% for training, and the remaining for validation. This yields two multi-task regression datasets, one for pre-training and the other for adaptation. A sample from the pre-training dataset comprises a 5D input vector $\mathbf{x} \in [0, 1]^5$ and the scalar regression targets $f_1(\mathbf{x}), ..., f_{20}(\mathbf{x})$.

**Method.** We pre-train a Neural Interpreter to regress all 20 functions simultaneously. To do this, we feed it a set of 25 elements as input: the first 5 are the components of $\mathbf{x}$ (with learnable positional embeddings [44] added in) and the remaining 20 are learned CLS tokens (vectors), one for each function. We attach a (shared) regression head to the output elements corresponding to the CLS tokens, and train their outputs to regress the respective function. Having pre-trained the model for 20 epochs, we finetune it for an additional 3 epochs on the 10 reserved functions $\{f_{21}, ..., f_{30}\}$. For the latter, we always instantiate and train 10 new CLS tokens, corresponding to the new functions; however, we

Table 1: Mean Coefficient of Determination ($R^2$) and StdDev over 10 tasks after training various sets of parameters. **Gist:** By just allowing the functions to *rewire* themselves by training only the type-matching parameters, one approaches the performance of tuning all remaining parameters, including those of the functions themselves. This suggests that the functions have learned recomposable primitives.

| Parameter Set | $R^2$ |
|---|---|
| All Parameters (Pretraining) | $0.9983 \pm 0.0005$ |
| Finetuning CLS Tokens | $0.9202 \pm 0.0198$ |
| + Type Inference Parameters | $0.9857 \pm 0.0034$ |
| + Remaining Parameters | $0.9953 \pm 0.0013$ |

investigate three settings in which three different sets of parameters remain frozen. In the first setting, all parameters remain frozen, implying that the CLS tokens are the only trainable parameters. In the

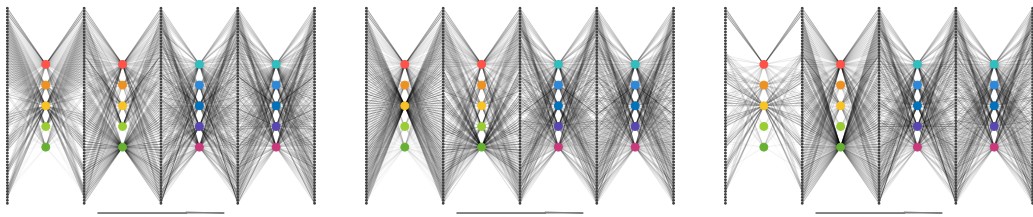

Figure 3: Visualization of computational paths taken by input set elements corresponding to three different samples as they progress through a Neural Interpreter. Colored dots identify functions (same color implies shared parameters), and the weight of the lines denote their compatibility with set elements. **Gist:** There are variations (but also similarities) between samples in how their constituent set elements are routed through the network.

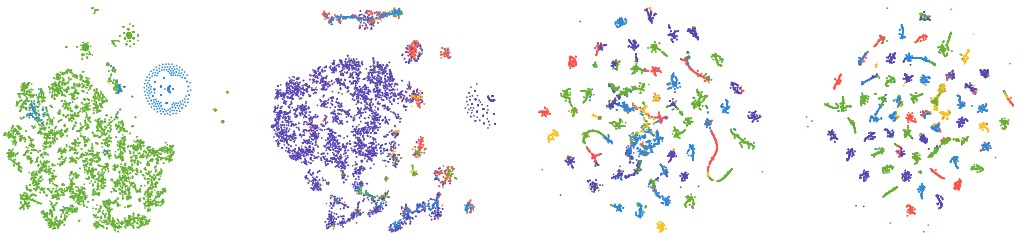

Figure 4: t-SNE embeddings of the inferred types of set elements as they progress through a Neural Interpreter with two scripts with two function iterations each. The color identifies the closest function in type space, and the progression from left to right is over the function iterations. **Gist:** Types are more clustered in the later function iterations, suggesting that the input set elements gradually *develop* a type as they progress through the network.

second setting, we unfreeze the parameters of the type matching mechanism (function signatures and parameters of the type inference MLP, in addition the CLS tokens). In the third setting, we unfreeze all parameters and finetune the entire model. Additional details in Appendix B.

**Hypothesis.** By finetuning just the type-matching parameters, we only permit adaptation in how information is routed through the network. In other words, we only allow the computational units to *rewire* themselves in order to adapt to the new task at hand, while preserving their functionality. Now if the computational primitives that are learned during pre-training are recomposable, one would expect the performance of Neural Interpreters having finetuned just the type-matching parameters to approach that obtained by finetuning all the parameters, the latter including ones that determine the functionality of the computational primitives.

**Results.** Table 1 compares the coefficients of determination[1] ($R^2$) obtained in each of the investigated finetuning settings. We find that relative to finetuning just the CLS tokens, the performance difference between finetuning all parameters and just the type matching parameters is small. This is in line with expectation, suggesting that Neural Interpreters are indeed capable of learning recomposable computational primitives.

### 4.2 Multi-Task Image Classification

In this section, we evaluate Neural Interpreters in a multi-task image classification setting. Our goals are (a) to determine whether the inductive bias helps with fast adaptation, (b) to investigate whether the interpreter can indeed function as a general instruction executor, as intuited in Section 2, and (c) to demonstrate that the proposed architecture produces modules that can function autonomously (without additional training objectives that encourage this behaviour). Additional results in Appendix C analyze the effects of varying hyper-parameters.

**Task Definition.** We consider three related datasets sharing the same label semantics, namely SVHN [41], MNISTM [20] and MNIST [35]. The images therein are upsampled to shape $32 \times 32$ (if required), and the resulting image is augmented with RandAugment [14]. Subsequently, the augmented images are split [13] to 64 patches of shape $4 \times 4$. In addition, we also use unaugmented samples from the K-MNIST dataset [12] of Hiragana characters to probe fast-adaptation to new data.

---

[1] $R^2 = 1$ implies perfect fit; a model that regresses to the mean has $R^2 = 0$.

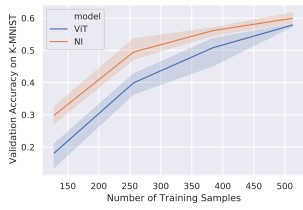
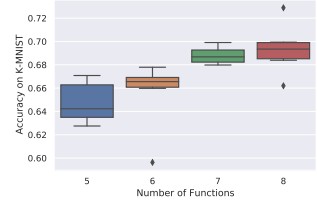
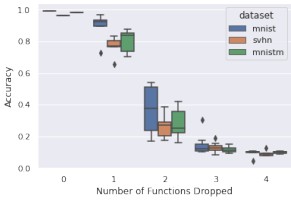

|(a) Fast adaptation|(b) Capacity extension|(c) Dropped functions|

Figure 5: *Figure 5a:* Performance of Neural Interpreters (NI) compared to that of a Vision Transformer (ViT) that performs equally well on the validation set (y-axis), plotted against the number of training samples presented (x-axis). We see Neural Interpreters can adapt faster in the low-data regime. *Figure 5b:* Validation performance (y-axis) of a Neural Interpreter trained with 5 functions but finetuned with various number of functions (x-axis). We see that performance increases with increasing functions, showing that the model does not *overfit* to the number of functions it was trained with. *Figure 5c:* Accuracy on the digits dataset as a function of the number of dropped functions. We find that the performance degrades gracefully as functions are dropped, suggesting that learned functions are autonomous, i.e. they can operate in the absence of other functions.

**Method.** We train Neural Interpreters for 100 epochs on the combined *digits* dataset described above. The input set contains 67 vector valued elements – the first 64 corresponding to linear embeddings of the $4 \times 4$ patches, and the remaining 3 to learnable CLS tokens (vectors), one for each dataset. For each CLS token, a linear classification head is attached to the corresponding output; given an input sample from a certain dataset, the respective classification head is trained to predict the correct class. To inject position information into the model, we use a variant of the relative positional encoding scheme described in [13], but applied only to the first 64 input elements (corresponding to the image patches). Having pre-trained on the digits dataset, we finetune the model on K-MNIST with varying numbers of samples for 10 epochs. For additional details, please refer to Appendix C.

**Baseline.** Vision Transformers (ViT) [13, 17] make the natural baseline for Neural Interpreters, given the fact that the former is a special case of the latter. The set-up with CLS tokens and classification heads is identical to that of Neural Interpreters, as is the training protocol. We use a light-weight 8-layer deep model with approximately 1.8M parameters, but ensure that the considered Neural Interpreter model is even lighter, with roughly 0.6M parameters.

**Hypotheses. (1)** It has been suggested that a model that appropriately modularizes the learning problem should excel at fast-adaptation to a new but related data distribution [6]. If Neural Interpreters obtain a useful modularization of the problem, we should expect it to be able to adapt faster to the K-MNIST dataset than a non-modular baseline like the ViT. This is because some of the visual primitives present in the digit datasets (e.g., motifs, strokes or shapes) might reoccur in the K-MNIST dataset, which Neural Interpreters should be able to leverage. **(2)** Recall that in Section 2, we positioned the interpreter as a general purpose instruction executor, capable of being programmed by a code vector. If this is indeed the case, and if the interpreter does not overfit to the functions it was trained with, we should expect the capacity (and therefore performance) to increase as we add and train new functions (i.e., tuples of code and signatures), while keeping all other parameters fixed. **(3)** If functions (modules) are indeed autonomous, we should expect that as some of them are removed at test time, the others are still able to maintain performance. Consequently, the accuracy of the Neural Interpreter (trained with all functions) should only degrade gracefully as functions are (randomly) removed at test time.

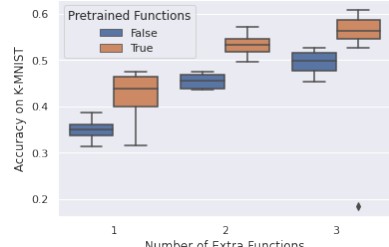

Figure 6: Performance on K-MNIST after adaptation. In one case (orange), functions pretrained from the digits dataset are kept intact. In another case (blue), function codes and signatures are randomly sampled before adaptation. We find that the former performs better. **Gist:** Knowledge acquired during pretraining is indeed reused during adaptation. This is apparent when comparing with a baseline where this knowledge is destroyed, leading to decreased performance after adaptation.

**Results.** Figure 5a compares the fast-adaptation performance of a Vision Transformer to that of a Neural Interpreter. While both models achieve almost identical performance on the validation set for all datasets (the difference being less than 0.2% in favor of Vision Transformers), we find that Neural Interpreters significantly outperform Vision Transformers at fast-adaptation in the low-data

regime, and the performance gap is only gradually closed as more data becomes available. This observation supports hypothesis (**1**). Further, in Figure 6, we establish a baseline where the functions pretrained on the digits dataset are destroyed prior to adaptation. We find that keeping the functions acquired from pretraining intact leads to significantly improved adaptation performance, suggesting that knowledge acquired during pretraining is indeed being reused during adaptation. This lends further support to hypothesis (**1**). Figure 5b shows the validation performance of a Neural Interpreter that was pre-trained with 5 functions on the digits dataset, but tested on the K-MNIST dataset with varying numbers of functions, having finetuned just the function signatures and codes. We find that the performance improves as new functions are added at adaptation-time, supporting hypothesis (**2**). Figure 5c shows the effect of randomly removing functions on the model accuracy. We observe that the performance on all datasets degrades gracefully as more functions are removed, thereby supporting hypothesis (**3**). In addition, Figure 3 visualizes the computational path taken by input set elements as they progress through the network, verifying that there is diversity in how samples are routed through the network, and shows that the routing mechanism discriminates between samples. Figure 4 shows that the input set elements gradually *develop* a type as they progress through the model. Additional figures in Appendix C analyze the effect of varying the number of scripts, number of function iterations, number of functions, kernel truncation parameter, dimension of type space $\mathcal{T}$ and freezing the function codes / signatures on the validation performance. The key finding is that, on the one hand, a wide range of hyper-parameter settings leads to performant models; on the other hand, there are patterns in what hyper-parameters perform consistently well.

### 4.3 Abstract Reasoning

In this section, our goal is to (a) use visual abstract reasoning tasks to evaluate whether Neural Interpreters are capable of systematic (compositional) generalization, and (b) characterize how Neural Interpreters maintain performance when the amount of compute is reduced at test time.

**Task Definition.** Progressively Generated Matrices (PGM) [5] is a procedurally generated series of datasets based on Raven Progressive Matrices (RPMs) [47], a popular IQ test designed for humans. The datasets, each comprising around 1.2M samples, aim to assess abstract and analogical reasoning capabilities of machine learning models beyond surface statistical regularities [29, 45]. Each sample in a dataset (Figure 7) comprises 8 context panels arranged in an incomplete $3 \times 3$ matrix, and 8 candidate answer panels (a panel being an $80 \times 80$ image). Given access to the context panels, the model must predict which of the candidate images is the most appropriate choice to complete the matrix. The content of the panels in a matrix are related by one or more *triples*, where a triple comprises a logical rule ($\in$ progression, XOR, OR, AND, consistent union) applied to an attribute ($\in$ size, type, color, position, number) of an object of a type ($\in$ shapes, lines). There are 8 datasets in the series, of which 7 measure systematic generalization in different ways, i.e., the test datasets contain samples where the panels are related in ways that are not encountered during training. In this work, we consider 6 datasets that probe the compositional reasoning ability of the model, namely: Interpolation, Extrapolation, Held-out (H.O.) triples, H.O. pairs of triples, H.O. Attribute Pairs, and Neutral. We omit H.O. line-type and H.O. shape-type datasets, since they stress the convolutional feature extraction instead of compositional reasoning component of the model. Please refer to Appendix D and [5] for additional details.

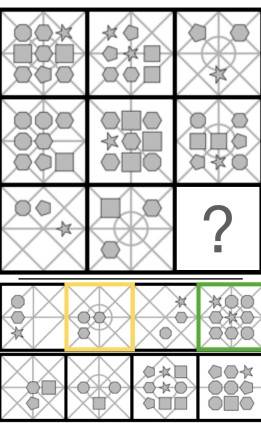

Figure 7: A PGM task. **Top:** Context panels. **Bottom:** Candidate panels. WReN and ViT predict the wrong answer (yellow), whereas NI predicts the correct one (green).

**Method.** Each panel (context and choice) is embedded with a shallow convolutional neural network to obtain an embedding vector. The input to the model is a set of 10 elements, comprising the embeddings of 8 context panels, that of a single candidate panel, and a CLS token (learnable vector). The model output corresponding to the CLS token is fed as input to the prediction head, which is trained to output a score measuring the compatibility of the candidate panel to the context panels. The final prediction is obtained by applying a softmax over the scores of all candidate panels. We note that this set-up resembles the one proposed in [5] for WReNs, and refer to Appendix D for details.

**Baselines.** We train Vision Transformers [13, 17] and Neural Interpreters with the same architectural scaffolding (described above). Additional baselines include Wild Relational Networks (WReN) [5],

Multiplex Graph Networks (MXGNet) [59] and WReNs atop disentangled representations (VAE-WReN) [54]. Regarding the latter, we note that disentangled representations only improves on the feature extraction component of the WReN model; as such, it is entirely complementary to our contribution, which is the abstract reasoning component. Future work may explore replacing the convolutional embeddings with disentangled representations.

**Hypothesis.** Recall that the test sets of (all but one) PGM datasets contain samples where the panels are not related in a way that is encountered during training. A model that is able to infer and reason about the underlying compositional structures is therefore more likely to be successful. If the inductive biases do indeed enable Neural Interpreters to factorize knowledge in to recomposable primitives, we might expect them to be able to dynamically recombine these primitives in order to generalize to problems it has not encountered during training and thereby excel at this task.

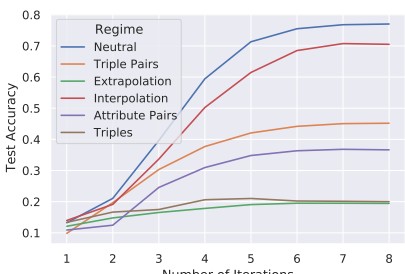

Figure 8: Test performance of a Neural Interpreter as a function of the number of function iterations (i.e. the amount of inference-time compute). **Gist:** Neural Interpreters are capable of trading-off performance with compute.

**Results.** Table 2 tabulates the validation (in-distribution) and test (out-of-distribution) accuracies obtained by all models for the various considered datasets. We make two observations: (1) Neural Interpreters are competitive in terms of test accuracy, outperforming both Vision Transformers and the prior state-of-the art in 4 of 6 datasets. (2) Vision Transformers are competitive in-distribution: they outperform the state of the art (excluding Neural Interpreters) in 5 of 6 datasets in terms of validation accuracy. However, they are outperformed in terms of test accuracy by either Neural Interpreters or other baselines in all datasets. In addition, Figure 8 shows how the test-set performance varies with the number of function iterations, for a model that was trained with a fixed number of function iterations (8). We note that the number of function iterations is proportional to the amount of computation used by the model. In all cases, the model maintains at least 70% of its original performance while using only half the amount of compute; in most cases, at least 80% of the original performance is maintained. This suggests that Neural Interpreters can function as anytime predictors [63], which are models that can trade-off performance with compute.

Table 2: Performance (prediction accuracy) of all models on different generalization regimes of PGM datasets. Note that the *Val.* performance measures in-distribution generalization, and the *Test* performance measures out-of-distribution generalization on the corresponding datasets (except for Neutral). *Extra.* refers to Extrapolation, *Attribute P.* to Attribute Pairs and *Triple P.* to Triple Pairs.

| Regime Model | Neutral | | Interpolation | | Attribute P. | | Triple P. | | Triples | | Extra. | |
|---|---|---|---|---|---|---|---|---|---|---|---|---|
| | Val. | Test | Val. | Test | Val. | Test | Val. | Test | Val. | Test | Val. | Test |
| WReN [5] | 63.0 | 62.6 | 79.0 | 64.4 | 46.7 | 27.2 | 63.9 | 41.9 | 63.4 | 19.0 | 69.3 | 17.2 |
| VAE-WReN [54] | 64.8 | 64.2 | - | - | **70.1** | **36.8** | 64.6 | 43.6 | 59.5 | **24.6** | - | - |
| MXGNet [59] | 67.1 | 66.7 | 74.2 | 65.4 | 68.3 | 33.6 | 67.1 | 43.3 | 63.7 | 19.9 | 69.1 | 18.9 |
| ViT | 73.3 | 72.7 | **89.9** | 67.7 | 69.4 | 34.1 | 67.6 | 44.1 | 73.8 | 15.9 | **92.2** | 16.4 |
| NI (ours) | **77.3** | **77.0** | 87.9 | **70.5** | 69.5 | 36.6 | **68.6** | **45.2** | 79.9 | 20.0 | 91.8 | **19.4** |

# 5 Conclusion

We have presented Neural Interpreters, a self-attention based architecture capable of learning a system of recomposable computational primitives. Neural Interpreters relax the rigidity in how computation is reused in current neural architectures, and our experiments show that the modular inductive biases it incorporates facilitate systematic generalization and sample-efficient adaptation to new tasks.

There are multiple exciting avenues of future research. One line of work could leverage the capacity extension capability of Neural Interpreters in a continual learning setting, thereby enabling *continuous integration* of knowledge in a neural model. Another promising direction entails using Neural Interpreters as a backbone for learning world models, where systematic generalization and out-of-distribution robustness are of paramount importance.

## Acknowledgements

The authors would like to thank Sharath Chandra Raparthy and Sarthak Mittal for providing valuable feedback on the draft. This work was in part supported by the German Federal Ministry of Education and Research (BMBF) through the Tübingen AI Center (FKZ: 01IS18039B). YB acknowedges support from the Canada CIFAR AI Chair Program, and Samsung Electronics Co., Ltd.

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
