# A  General Details about the Architecture

## A.1  Hyperparameters and How to Set Them

Neural Interpeters introduce a number of components that are not present in Vision Transformers, and accordingly, it introduces additional hyperparameters. While we found a large range of hyperparameters can work well in practice, there are patterns that warrant discussion. In this section, we provide a detailed discussion of these hyperparameters, and how these are set in this work. We also remark that there might be other hyperparameter settings that work well for different settings, and the insights in this section should merely function as a guide.

### A.1.1  Partitioning Depth between LOCs, Function Iterations and Script

There are three distinct ways of increasing the depth of Neural Interpreters.

1. **Increasing the number of Function Iterations.** Increasing $n_i$, the number of function iterations, is a natural way of increasing the depth of Neural Interpreters *without* increasing the number of parameters. Large $n_i$ encourages the recursive and iterative reuse of computation [15], but might result in a scarcity of parameters.

2. **Increasing the number of Scripts.** Increasing $n_s$, the number of scripts, is a way of increasing the depth of Neural Interpreters *while* increasing the number of parameters. Larger $n_s$ tends to result in models that are easier to train, potentially due to a larger number of good solutions in the parameter space [11] (owing to the larger number of parameters). However, if $n_s$ is increased at the price of decreasing $n_i$, one might forego some inductive bias towards iterative reuse of computation.

3. **Increasing the number of LOCs.** Increasing $n_l$, the number of LOCs, also increases the depth *while* increasing the number of parameters. However, unlike increasing $n_s$, increasing $n_l$ results in a deeper block of layers being recursively applied. Increasing $n_l$ might come at the price of decreasing $n_i$, in which case some recursive inductive bias is foregone; or it might come at the price of decreasing $n_s$, which might result in models that are less consistent.

**Recommendation.** If training is less stable or in-distribution performance is important, one should consider increasing the number of scripts $n_s$. If the training is stable but out-of-distribution generalization or fast-adaptation performance is important for the application, one should consider increasing the number of function iterations $n_i$. If there is additional budget for hyper-parameter search, one could consider tuning the number of LOCs (starting with $n_l \in \{1, 2\}$).

### A.1.2  Increasing the Number of Functions

Increasing $n_f$, the number of functions, is a parameter efficient way of increasing the width of the network in a model-parallelizable way. This is especially apparent from Equation 9, where the index over functions ($u$) can be effectively folded in to the batch-axis. Further, we found benefits in increasing the number of functions (also in terms of in-distribution performance), suggesting that the distribution of parameters between interpreter and the codes (as described in Equation 4) is scalable.

**Recommendation.** The number of functions can be safely increased to match available resource capacity.

### A.1.3  Kernel Truncation Parameter and Dimension of Type Space

These hyperparameters (inherited from [46]) have to do with routing of information through the network. The truncation parameter $\tau \in [0, 2)$ controls the hardness of the routing – if $\tau$ is small, functions are only granted access to set elements whose types lie in the immediate vicinity of their signatures. For larger $\tau$, functions may be granted access to set elements whose types are less similar to their signatures in type-space, albeit the said elements are down-weighted by the kernel. The type space dimension $d_{\text{type}}$ controls the amount of flexibility afforded to the routing mechanism. Intuitively, larger $d_{\text{type}}$ implies that there are more ways to how the signature and type vectors can be positioned in the type space $\mathcal{T}$ (a hypersphere of dimension $d_{\text{type}}$) relative to each other.

**Recommendation.** These hyperparameters may vary with the problem at hand. If sparsity is desired, one should consider lower values for $\tau$. If training is less stable, larger values of $\tau$ might mitigate the

issue. We find $\tau \in [1.2, 1.7]$ to be a reasonable range for hyperparameter sweeps. As for $d_{\text{type}}$, we find all values between 20 and 50 to work well in our experiments.

### A.1.4 Learning Function Signatures and Code

When pre-training the model, one decision that must be made is whether or not the function signatures and codes should be trained. Note that freezing these parameters at the pre-training stage does not necessarily constrain the model in a significant way – if the function signatures are fixed, the type-inference MLP can adapt (Equation 2); likewise, if function codes are frozen, the weight matrices $W_c$ (Equation 9) can adapt. Note that this applies in the pre-training phase, where the type inference MLP and the interpreter parameters are allowed to adapt.

**Recommendation.** While we did not find a large difference, runs with frozen function codes were slightly less consistent than the ones with learned function codes. At the same time, runs with frozen function signatures tended to perform at least as well as the ones that learned function signatures, if not slightly better.[2]

### A.1.5 Choice of an Optimizer and Scheduler

Like for most self-attention based models (including the transformer [58]), the choice of an optimizer and learning rate schedule plays an important role. A common practice is to use Adam with a linear learning rate warm-up and cosine annealing (once per optimization step). However, learning rate warm-up is known to be a heuristic to control the variance of Adam learning rate in the early stages of training, a problem that Rectified Adam [36] (RAdam) solves in a more principled way while eliminating a sensitive hyperparameter (the number of warm-up steps). Further, for certain adaptation tasks where the loss-landscape can potentially be challenging, we found Shampoo [25] to work particularly well.

**Recommendation.** For pre-training Neural Interpreters, we can recommend the RAdam optimizer with a cosine annealing schedule (without warm-up). We anneal the learning rate by $\sim$two orders of magnitude over 80-90% of the training steps, and keep the learning rate at the minimum for the remainder of the steps. While we found RAdam to also work well for most finetuning experiments, Shampoo [25] with appropriately tuned learning rate can serve as a reasonable alternative in the event that RAdam does not perform as expected.

## A.2 Limitations and Future Work

**Lack of Top-Down Processing.** Like most self-attention based models, Neural Interpreters process their inputs in a bottom-up manner where the input is encountered only once, which is at the first (input) layer. However, top-down processing of information is known to be a useful prior [24, 28, 37], and future work may explore incorporating this in the proposed architecture.

**Higher Order Functions.** While Neural Interpreters are a step towards models that can flexibly compose computational primitives, they are (in their current form) missing certain notions from functional programming that could potentially serve as useful inductive biases. One of these notions is that of higher-order functions, i.e., functions that can manipulate other functions depending on the context.[3] Support for higher-order functions can enable the model to no longer rely on a discrete set of pre-learned functions; instead, the model can learn to create new functions on the fly (i.e. at test time). We nevertheless note that Neural Interpreters already posses some features that can facilitate this inductive bias, the most important one being explicit representation of function codes (which can be manipulated by functions, just like other set elements).

**Lazy Function Execution.** Another potentially powerful notion (borrowed from functional programming) that is yet to be incorporated is that of *lazy* or *deferred* function executions. When implemented, this can enable models that support functional representations, or *abstractions* as they are known in the language of lambda calculus.

---

[2]This is less surprising in light of the fact that the type-inference MLP has a larger number of parameters that can be adapted during training.

[3]In python, these are like decorators.

## A.3 Broader Impact

Neural Interpreters is a general neural back-bone that can be used in a variety of applications that we may not yet foresee. Nevertheless, the presented work provides an architecture that is computationally scalable, implying that a user might be tempted to experiment with more compute than is strictly necessary for obtaining good results; if not powered by nuclear or other renewable sources of energy, this might result in a larger carbon footprint.

## B Learning Fuzzy Boolean Expressions

### B.1 Sampling Fuzzy Boolean Functions

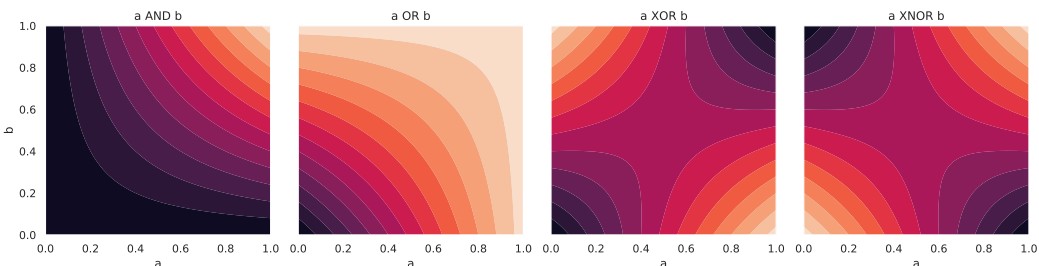

Figure 9: Visualization of fuzzy relaxations of binary operations mapping $a \in [0, 1]$ and $b \in [0, 1]$ to a value in $[0, 1]$. From left to right: and, or, xor and xnor.

In what follows, we define a family of smooth functions mapping from the unit hyper-cube $[0, 1]^N$ to $[0, 1]$. To this end, consider again the primitives defined in Equation 15. Where $x_i, x_j \in [0, 1]$, we define:

$$\texttt{and}(x_i, x_j) = x_i x_j \tag{16}$$

$$\texttt{not}(x_i) = \bar{x}_i = 1 - x_i \tag{17}$$

$$\texttt{or}(x_i, x_j) = x_i \oplus x_j = 1 - (1 - x_i)(1 - x_j) \tag{18}$$

Observe that if $x_i, x_j \in \{0, 1\}$, these operations reduce to their Boolean namesakes, and Equation 18 is consistent with de Morgan's law. In this sense, the primitives described above induce a *relaxation* of Boolean logic to real numbers on the compact interval $[0, 1]$. We note that this relaxation, called product fuzzy logic, is not unique: there exist other definitions of the and and not operations that define other logics (examples being Łukasiewicz and Gödel-Dummett logics).

Given these primitives, it is now possible to construct functions that resemble boolean functions in the cannonical disjunctive normal form (i.e., in the sum-of-products form). As an example, consider a vector $\mathbf{x} \in [0, 1]^5$, whose components we call $a, b, c, d, e \in [0, 1]$. One may now define a function:

$$f(\mathbf{x}) : [0, 1]^5 \rightarrow [0, 1], \ (a, b, c, d, e) \mapsto \bar{a}bcd\bar{e} \oplus a\bar{b}c\bar{d}e \oplus ab\bar{c}d\bar{e} \tag{19}$$

If $a, b, c, d, e$ were to be boolean (i.e., $\in \{0, 1\}$), the function $f$ would have a truth table where $f = 1$ only if $a = 0, b = 1, c = 1, d = 1, e = 0$, or $a = 1, b = 0, c = 1, d = 0, e = 1$, or $a = 1, b = 1, c = 0, d = 1, e = 0$. Conversely, given this truth table, it is possible to reconstruct $f$ in the sum-of-products form described above.

The above fact makes randomly sampling a fuzzy boolean function like sampling from the Bernoulli distribution: for all combinations of possible values of $a, b, c, d, e \in \{0, 1\}^5$, we sample the value of a boolean function $f(a, b, c, d, e) \sim \text{Bernoulli}(0.5)$ in order to populate the truth-table of $f$. Given the randomly sampled truth table, we construct the expression for $f$ in the sum-of-product form. Finally, we interpret the boolean expression (mapping from $\{0, 1\}^5 \rightarrow \{0, 1\}$) as a fuzzy boolean expression mapping from $[0, 1]^5 \rightarrow [0, 1]$ using the corresponding primitives defined in Equation 16 et seq.

### B.2 Hyperparameters

Please refer to Table 3.

Table 3: Hyperparameters for results in Table 1 (Learning Fuzzy Boolean Expressions).

| Parameters | Values |
|---|---|
| Batch size | 128 |
| Pretraining epochs | 20 |
| Finetuning epochs | 3 |
| Dimension of code vector ($\mathbf{c}$) | 128 |
| Dimension of intermediate features | 128 |
| Number of scripts ($n_s$) | 2 |
| Number of function Iterations ($n_i$) | 2 |
| Number of LOCs ($n_l$) | 1 |
| Number of functions ($n_f$) | 4 |
| Number of heads per LOC | 1 |
| Number of features per LOC head | 32 |
| Type Inference MLP Depth | 2 |
| Type Inference MLP Width | 128 |
| Frozen Function Signatures | `False` |
| Frozen Function Codes | `False` |
| Truncation Parameter ($\tau$) | 1.6 |
| Type Space Dimension ($d_\text{type}$) | 24 |
| Optimizer | RAdam [36] |
| Adam: learning rate (pre-training) | 0.006 |
| Adam: learning rate (finetuning) | 0.05 |
| Adam: $\beta_1$ | 0.9 |
| Adam: $\beta_2$ | 0.999 |
| Adam: $\epsilon$ | 1e-8 |
| Learning rate scheduler | None |

# C Multi-Task Image Classification

## C.1 The Digits Dataset

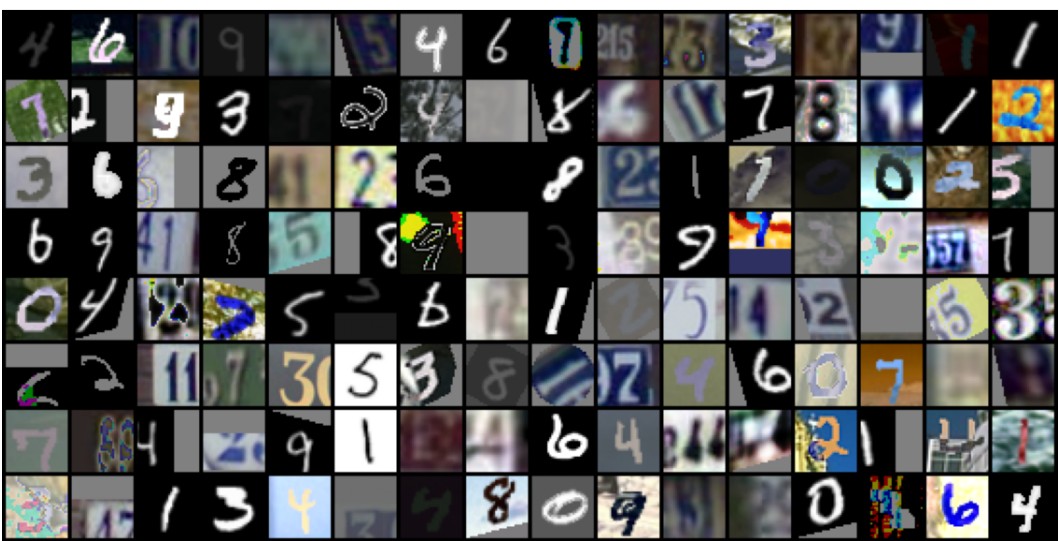

Figure 10: Augmented Samples from the Digits Dataset.

The Digits dataset is a concatenation of three datasets of labelled images of digits: SVHN [41], MNISTM [20], and MNIST [35]. All images are up-sampled to RGB images of size $32 \times 32$, and the combined training set has 193257 samples, whereas the validation set has 46032 samples. In addition

Table 4: Hyperparameters for pre-training the Neural Interpreter Model used in Figure 5.

| Parameters | Values |
|---|---|
| Batch size | 128 |
| Pre-training epochs | 100 |
| Dimension of code vector ($\mathbf{c}$) | 192 |
| Dimension of intermediate features | 192 |
| Number of scripts ($n_s$) | 1 |
| Number of function iterations ($n_i$) | 8 |
| Number of LOCs ($n_l$) | 1 |
| Number of functions ($n_f$) | 5 |
| Number of heads per LOC | 4 |
| Number of features per LOC head | 128 |
| Type Inference MLP Depth | 2 |
| Type Inference MLP Width | 192 |
| Frozen Function Signatures | True |
| Frozen Function Codes | False |
| Truncation Parameter ($\tau$) | 1.4 |
| Type Space Dimension ($d_{\text{type}}$) | 24 |
| Optimizer | RAdam [36] |
| Adam: $\beta_1$ | 0.9 |
| Adam: $\beta_2$ | 0.999 |
| Adam: $\epsilon$ | 1e-8 |
| Learning rate scheduler | Cosine (no warm-up) |
| Scheduler: $\eta_{\text{max}}$ (Max LR) | 0.0008 |
| Scheduler: $\eta_{\text{min}}$ (Min LR) | 0.000001 |
| Scheduler: Number of decay steps | 120000 |
| Number of parameters | $6.43 \times 10^5$ |
| Accuracy on SVHN | 96.2 % |
| Accuracy on MNISTM | 98.4 % |
| Accuracy on MNIST | 99.4 % |

to the images and labels, we also preserve information about which of the constituent datasets a sampled image originates from.

We use RandAugment [14] to augment the input images before feeding them to the model, and use the implementation from Pytorch Image Models [60]. Figure 10 shows augmented samples from the dataset.

## C.2 Hyperparameters

### C.2.1 Pre-training

Table 4 shows the hyperparameters used for pre-training the Neural Interpreter model considered in Figure 5. Table 5 shows the same, but for the Vision Transformer model.

### C.2.2 Finetuning

Both models shown in Figure 5 (top) were fine-tuned for 10 epochs with varying number of samples. We used the same batch-size as in pre-training (128). The error bands are with respect to 6 random seeds, where the random seed also determines the subset of K-MNIST that was used. We used RAdam optimizer with a constant learning rate, which was found with a grid search (0.03 for ViT and 0.05 for NI).

For the results shown in Figure 5 (bottom), the function codes and signatures were trained for 10 epochs on 8192 samples with Shampoo [25]. We again used 6 random seeds, and for each set of trainable parameters, we grid-searched the learning rate. We did not see good performance

Table 5: Hyperparameters for pre-training the ViT Model used in Figure 5.

| Parameters | Values |
|---|---|
| Batch size | 128 |
| Pretraining epochs | 100 |
| Dimension of intermediate features | 192 |
| Number of MLP Features | 192 |
| Depth | 8 |
| Number of heads | 3 |
| Number of features per head | 64 |
| Optimizer | RAdam [36] |
| Adam: $\beta_1$ | 0.9 |
| Adam: $\beta_2$ | 0.999 |
| Adam: $\epsilon$ | 1e-8 |
| Learning rate scheduler | Cosine (no warm-up) |
| Scheduler: $\eta_{\max}$ (Max LR) | 0.0008 |
| Scheduler: $\eta_{\min}$ (Min LR) | 0.000001 |
| Scheduler: Number of decay steps | 120000 |
| Number of parameters | $1.80 \times 10^6$ |
| Accuracy on SVHN | 96.3 % |
| Accuracy on MNISTM | 98.3 % |
| Accuracy on MNIST | 99.6 % |

with RAdam in this particular setting, suggesting that the loss landscape might necessitate the pre-conditioning that is present in Shampoo (but not in RAdam).

### C.2.3 Positional Encoding

We use a variant of the relative positional encoding scheme presented in [13], which we now describe. Consider an array $\mathbf{X}$ of shape $C \times H \times W$, where $C$ is the number of channels, and $H$ and $W$ can be interpreted as height and width. Note that the array $\mathbf{X}$ need not be an image; it could (for instance) be a collection of embedding vectors of patches, i.e., $\mathbf{X}_{\cdot,ij}$ could be the embedding vector (of dimension $C$) of the patch that is $i$-th from top and $j$-th from left.

We now denote with $\mathbf{e}_{\mathrm{row}}[i_2 - i_1]$ a vector that is a learned embedding of the difference of row-indices $i_2$ and $i_1$. Likewise, we let $\mathbf{e}_{\mathrm{col}}[j_2 - j_1]$ be a learned embedding of the difference of column-indices $j_2$ and $j_1$. Where $h$ indexes attention heads, we define:

$$e_{\mathrm{row}}^h[i_2 - i_1] = \mathbf{w}_{\mathrm{row}}^h \cdot \mathbf{e}_{\mathrm{row}}[i_2 - i_1] \tag{20}$$

$$e_{\mathrm{col}}^h[j_2 - j_1] = \mathbf{w}_{\mathrm{col}}^h \cdot \mathbf{e}_{\mathrm{col}}[j_2 - j_1] \tag{21}$$

Here, $\mathbf{w}_{\mathrm{row}}^h$ and $\mathbf{w}_{\mathrm{row}}^h$ are learned weight vectors, and $e_{\mathrm{row}}^h[i_2 - i_1]$ and $e_{\mathrm{col}}^h[i_2 - i_1]$ are scalars, one per attention head. This set-up allows each attention head to develop a positional bias independently from other heads, a feature we inherit from [13]. In the context of Neural Interpreters, we additionally allow functions to have their own positional bias, conditioned on its code $\mathbf{c}_u$. We have:

$$p_{\mathrm{row}}^{uh}[i_2 - i_1] = \mathrm{ModLin}_{\mathrm{row}}^h(\mathbf{e}_{\mathrm{row}}[i_2 - i_1]; \mathbf{c}_u) \tag{22}$$

$$p_{\mathrm{col}}^{uh}[j_2 - j_1] = \mathrm{ModLin}_{\mathrm{col}}^h(\mathbf{e}_{\mathrm{col}}[j_2 - j_1]; \mathbf{c}_u) \tag{23}$$

Here, $p_{\mathrm{row}}^{uh}[i_2 - i_1]$ and $p_{\mathrm{col}}^{uh}[j_2 - j_1]$ are scalars specific to function $u$ and attention head $h$. Finally, the overall positional bias is given as following, where broadcasting operations are implied:

$$b^{uh}[i_2 - i_1, j_2 - j_1] = (p_{\mathrm{row}}^{uh}[i_2 - i_1] + e_{\mathrm{row}}^h[i_2 - i_1]) + (p_{\mathrm{col}}^{uh}[j_2 - j_1] + e_{\mathrm{col}}^h[j_2 - j_1]) \tag{24}$$

Here, $b^{uh}[i_2 - i_1, j_2 - j_1]$ is the positional bias that is added to the pre-softmax dot-product attention weights coupling the embedding vectors $\mathbf{X}_{\cdot,i_1 j_1}$ and $\mathbf{X}_{\cdot,i_2 j_2}$ at function $f_u$ and attention head at index $h$. We remark that this scheme only differs from [13] in that we allow each function to develop its own positional bias.

### C.3 Additional Results and Ablations

In order to understand the effect of various hyperparameters, we analyze the results of a random sweep over 100 runs on the Digits dataset. The distributions over sweep parameters are presented in Table 6.

Table 6: Distribution over hyperparameters used in the sweep. $\mathcal{U}$ denotes the uniform distribution.

| Parameters | Distribution |
| --- | --- |
| Truncation parameter ($\tau$) | $\mathcal{U}([0.7, 1.7])$ |
| Dimension of type space ($d_{\text{type}}$) | $\mathcal{U}(\{4 * i \mid i \in \{2, 3, ..., 12\}\})$ |
| Number of functions ($n_f$) | $\mathcal{U}(\{1, 2, 3, 4, 5\})$ |
| Num. of scripts, function iterations, and LOCs ($n_s, n_i, n_l$) | $\mathcal{U}(\{(2, 2, 2), (2, 4, 1), (4, 2, 1), (1, 8, 1)\})$ |
| Frozen function signatures | $\mathcal{U}(\{\texttt{True}, \texttt{False}\})$ |
| Frozen function codes | $\mathcal{U}(\{\texttt{True}, \texttt{False}\})$ |
| Frozen patch embeddings [10, 56] | $\mathcal{U}(\{\texttt{True}, \texttt{False}\})$ |

**Kernel Truncation and Dimension of Type Space.** In Figure 11, we select for each dataset the top 10% of all runs (w.r.t. validation performance), and plot a Kernel Density Estimate of their type space dimensions ($d_{\text{type}}$) and truncation parameters ($\tau$). We find that while the optimal $\tau$ and $d_{\text{type}}$ only somewhat depend on each other, there are minor variations between the SVHN and MNIST-M vs. MNIST. We speculate that this is due to SVHN and MNIST-M having cluttered backgrounds; the flexibility afforded by a larger type space is less desirable when the model must learn to suppress background clutter by routing noisy patches through similar functions.

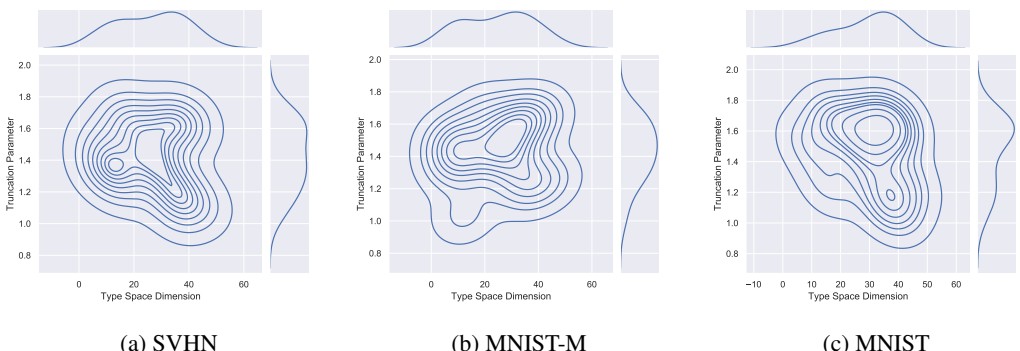

| (a) SVHN | (b) MNIST-M | (c) MNIST |

Figure 11: Kernel Density Estimates of truncation parameters $\tau$ and type space dimensions $d_{\text{type}}$ of the top 10% of runs for each dataset.

**Number of Functions.** Figure 12 shows the kernel density estimate of the validation performance of *all* 100 runs, conditioned on the number of functions. We read that on the one hand, runs with 5 functions perform consistently well; on the other hand, there exist runs with a single function that perform well, but most of these runs fail. This is consistent with the recommendation in Section A.1.2.

**Frozen Patch Embeddings, Function Signatures and Codes.** Figure 13 shows the validation performance of top 10% of runs (for the respective dataset), with or without frozen patch embeddings, function signatures and codes. As elaborated in Section A.1.4, it is not surprising that Neural Interpreters can work well even when function codes and signatures remain frozen during training. We find that freezing function signatures can be marginally beneficial, but freezing function codes less so. We also experimented with freezing the patch embeddings, as recommended in [10, 56], and find that it slightly improves performance.

**Number of Scripts, Function Iterations and LOCs.** Figure 16 shows the conditional kernel density estimates of validation performance, conditioned on the number of scripts. We again find that all evaluated configurations can work well. A larger number of scripts can stabilize training and lead to consistent in-distribution performance, as expected from Section A.1.1.

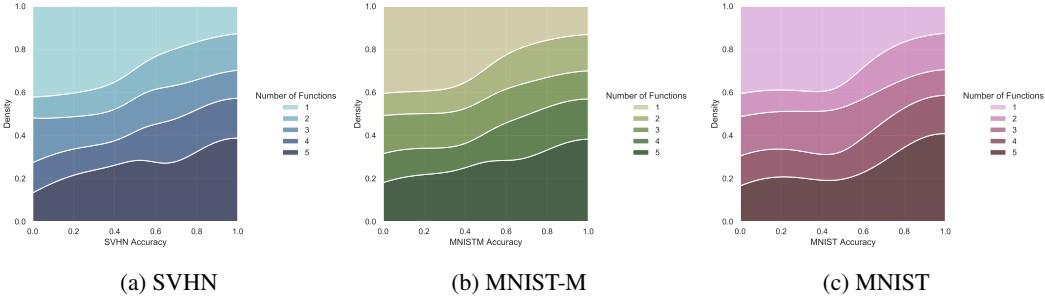

(a) SVHN        (b) MNIST-M        (c) MNIST

Figure 12: Conditional Kernel Density Estimates of validation performance, conditioned on the number of functions $n_f$.

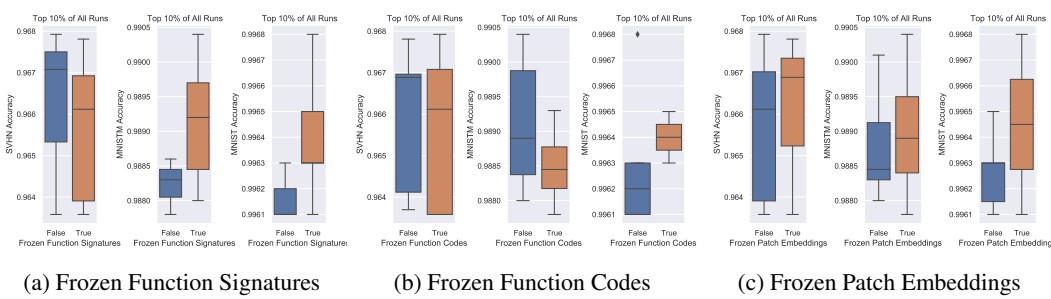

(a) Frozen Function Signatures    (b) Frozen Function Codes    (c) Frozen Patch Embeddings

Figure 13: Box plots of validation performance of top 10% of all runs.

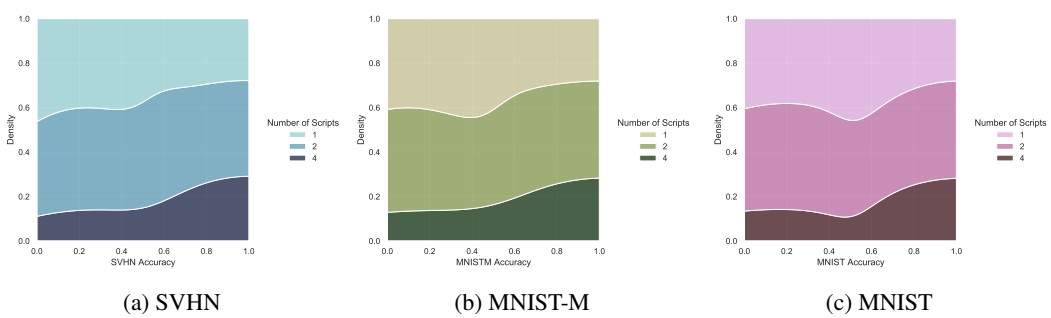

(a) SVHN        (b) MNIST-M        (c) MNIST

Figure 14: Conditional Kernel Density Estimates of validation performance, conditioned on the number of scripts $n_s$.

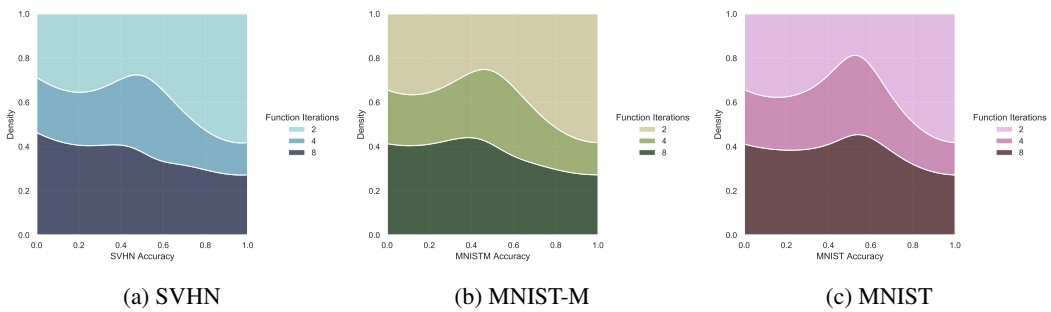

(a) SVHN        (b) MNIST-M        (c) MNIST

Figure 15: Conditional Kernel Density Estimates of validation performance, conditioned on the number of function iterations $n_i$.

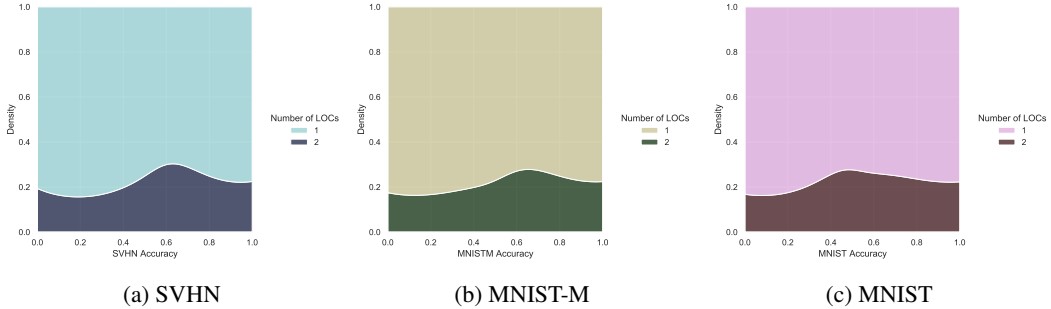

|  (a) SVHN | (b) MNIST-M | (c) MNIST |

Figure 16: Conditional Kernel Density Estimates of validation performance, conditioned on the number of LOCs $n_l$.

# D    Abstract Reasoning with PGMs

Progressively Generated Matrices (PGMs) [5] have been used as a diagnostic dataset to study the compositional generalization capability of machine learning models [54, 59]. The dataset consists of complex visual analogical reasoning tasks that require relational reasoning between attributes of different objects. The 'object types' include shape and line comprising of the 'attribute types': size, type, color, position, and number, where each attribute can take one of a finite number of discrete values. The 'relationship types' consist of progression, XOR, OR, AND, and consistent union. The structure of a PGM task is governed by *triples*, which are defined by applying a certain relationship type to the attributes of the objects. On an average, one to four relationships are used per task.

To study the various aspects of generalization in models, [5] introduced 8 different sub-datasets, corresponding to different generalization regimes of compositional reasoning. Except for the *Neutral* regime, the test dataset in each regime measures the out-of-distribution generalization i.e., the test and the training datasets are different in a clearly defined manner. We use 6 of such regimes in this work, namely: Interpolation, Extrapolation, Held-out (H.O.) triples, H.O. pairs of triples, H.O. Attribute Pairs, and Neutral. The details on these regimes are provided below.

### D.0.1    Generalization Regimes

**Neutral:** The neutral regime measures the in-distribution generalization, i.e., the training and the test sets consist of any triples.

**Interpolation:** In the training dataset of interpolation regime, the discrete values of the attributes are restricted to even numbers whereas the test set consists of odd-valued attributes.

**Extrapolation:** For the training dataset of extrapolation regime, the attribute values were restricted to the lower half of the discrete set whereas the test set consists of values sampled from the upper half of the discrete set.

**Held-out Triples:** The PGM dataset contains 29 unique triples. In the test set of held-out triples, 7 of such triples were held-out, while the rest of the triples are used to create the training dataset.

**Held-out Pairs of Triples:** All tasks contain at least two triples, leading to 400 viable triple pairs [5]. In *Held-out Pairs of Triples*, 360 such pairs are randomly allocated to the training dataset and rest to the test dataset.

**Held-out Attribute Pairs:** Here, each task consists of at least two triples, where there are 20 viable pairs of attributes. Of these 20 pairs, 16 have been used to create the training set while the remaining 4 are used in the test set.

### D.1    Details of PGM Experiments

For each PGM sub-dataset, we train multiple models for both Vision Transformers and Neural Interpreters. Each model is trained for 30 epochs and the model selection is done by evaluating its performance on validation datasets. The reported test accuracy in Table 2 corresponds to the best

Table 7: Hyperparameters and their values that are kept the same in all the PGM experiments.

| Parameters | Values |
|---|---|
| Batch size | 72 |
| Epochs | 30 |
| Dimension of code vector ($\mathbf{c}$) | 192 |
| Dimension of intermediate features | 192 |
| Number of scripts ($n_s$) | 2 |
| Number of function iterations ($n_i$) | 8 |
| Number of LOCs ($n_l$) | 1 |
| Number of functions ($n_f$) | 5 |
| Number of heads per LOC | 4 |
| Number of features per LOC head | 32 |
| Type Inference MLP Depth | 2 |
| Type Inference MLP Width | 192 |
| Variable Features dimensions | 192 |
| Frozen Function Codes | `False` |
| Optimizer | RAdam |
| Adam: beta1 | 0.9 |
| Adam: beta2 | 0.999 |
| Adam: epsilon | 1e-8 |
| Adam: learning rate | 0.0004 |
| Learning Rate Scheduler | Cosine |
| Cosine Scheduler Eta Max | 0.0004 |
| Cosine Scheduler Eta Min | 0.0001 |
| Number of parameters | 1.6M |

Table 8: Hyperparameters whose values are randomly sampled from the given ranges for each experiment.

| Parameters | Ranges |
|---|---|
| Kernel truncation parameter ($\tau$) | $[1.3, 1.7]$ |
| Type features dimensions | $[20, 24, 28, 32, 36, 40]$ |
| Detach Function Signatures | $[\texttt{True}, \texttt{False}]$ |
| Number of scripts ($n_s$) | $[1, 2, 4]$ |
| Function Iterations | $[4, 8, 16]$ |

validation performance. We perform hyper-parameter sweeps to find the best configuration of Neural Interpreters in each regime.

### D.1.1 Hyperparameter Settings

We perform random sweeps to find the optimal hyperparameters for each PGM regime. Due to the huge computational overload and the massive size of the datasets, the number of experiments in each sweep is limited to 35. We carried forward the knowledge that we learned from the digits experiments (Section 4.2), and perturbed only those hyperparameters that had significant influence on the model's performance. Apart from changing these selected hyperparameters, the models are identical in all aspects. Table 7 provides the hyperparameters that are kept the same in all the models, whereas Table 8 shows the hyperparameters that we perturb and the ranges from which their values are randomly sampled.

For the sake of consistency, we make sure that the number of computational steps remain the same in all the experiments. For PGM sweeps, the number of computational steps are set to be 16. We varied the numbers of scripts $n_s$ and function iterations $n_i$ such that their product comes out to be 16.

### D.1.2 Optimal Neural Interpreters Configuration for PGMs

After running the set of experiments, we found that the one configuration that outperformed all other configurations was with 2 scripts i.e., $n_s = 2$ and 8 function iterations. There are small fluctuations in the selection of kernel truncation parameter $\tau$ and dimensions of type space $d_{\text{type}}$ that we detail below in Table 9.

Table 9: Hyperparameters of Neural Interpreters for the considered PGM datasets.

| Regime | Neutral | Interpolation | Attribute P. | Triple P. | Triples | Extra. |
|---|---|---|---|---|---|---|
| Number of scripts ($n_s$) | 2 | 2 | 2 | 2 | 2 | 2 |
| Function iterations ($n_i$) | 8 | 8 | 8 | 8 | 8 | 8 |
| Kernel truncation parameter ($\tau$) | 1.62 | 1.62 | 1.40 | 1.66 | 1.42 | 1.42 |
| Type space dimensions ($d_{\text{type}}$) | 20 | 20 | 32 | 24 | 24 | 24 |
| Frozen function signatures | False | False | True | False | True | True |

## E  Diversity in Routing Mechanism

We further investigate whether the learned routing in neural interpreters is meaningfully diverse i.e. whether certain samples get routed through certain functions? To answer it, we visualize the t-SNE embeddings of the variable types in Figure 17. The color-codes represent the close affinities between variables and certain functions in type space. We compare it against the case where the routing is fixed at initialization Figure 18. It can be seen that in the randomly initialized routing the type-function assignments (given by the colors assigned to a dot) exhibit less structure and diversity, especially at the later function iterations. This suggests that the learning process in neural interpreters indeed induces non-trivial patterns in how information is routed between modules.

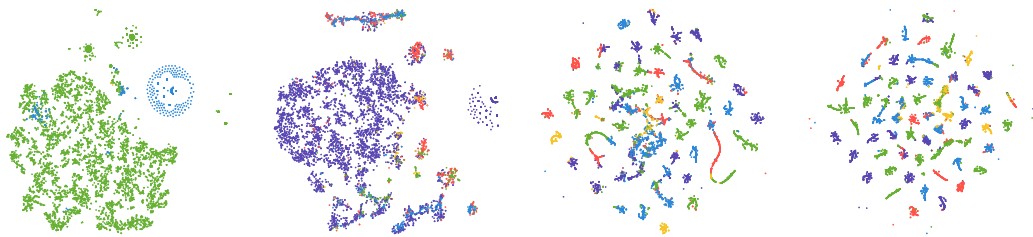

Figure 17: t-SNE embeddings of the inferred types of set elements as they progress through a Neural Interpreter with two scripts with two function iterations each. The color identifies the closest function in type space, and the progression from left to right is over the function iterations. **Gist:** Types are more clustered in the later function iterations, suggesting that the input set elements gradually *develop* a type as they progress through the network.

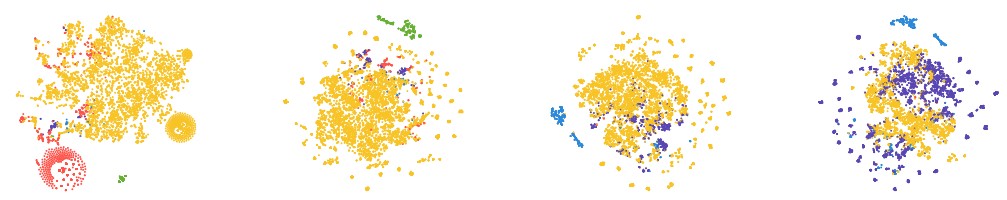

Figure 18: Same plot as above, but now with routing fixed at initialization. **Gist:** Inferred types of set elements exhibit less structure and diversity at initialization, especially at later function iterations. This suggests that the learning process indeed induces non-trivial patterns in how information is routed through the network.