# OpenReview forum: "Dynamic Inference with Neural Interpreters"
_NeurIPS.cc/2021/Conference — NeurIPS 2021 Poster_

### Official Review · Reviewer_zsVd · 2021-07-12

**Rating:** 6
**Confidence:** 4

**Summary:**

This paper proposes Neural Interpreter (NI), a drop-in replacement for Transformer layers that can be applied on set-valued inputs. The motivation behind the architecture is that common computation units in deep learning such as convolution layer is rigid in how they process the input, so to overcome this, NI leverages self-attention to dynamically determine computation path using its modular functions depending on the given input. The paper shows how NI performs well in transfer-learning and systematic generalization tasks.

**Limitations And Societal Impact:**

Yes

**Main Review:**

Originality & Significance
- Neural Interpreter’s way of dynamically composing functions can enable better generalization and its modularity give a way of expanding model - possibly continual learning although not studied in this paper. Similar to recent work Switch Transformer, NI proposes an efficient way of leveraging modular architecture with large number of parameters differing from other modular models such as SCOFF. Although promising, I find the experiments section not too strong as discussed in the next section.

Quality
- I like the model formulation proposed by NI of how it used functions to modulate and compose computations.
- Section 4.1 could have baseline models such as the plain transformer model to demonstrate if the inventions in NI is really making the task solvable.
- Additional experiments on continual learning datasets by continuously adding more functions to NI would strengthen the results.
- Analyzing or visualizing how modular functions are specialized would be beneficial. Figure 3 shows computation paths, but it is not easy to see what is going on. With a smaller number of functions on a simple task, it would be possible to clearly see if functions in NI can specialize according to the given input type. It would be also good to analyze how the truncation parameter can be used to control sparsity.
- Figure 5 (bottom) could also compare with ViT for example by adding more layers to ViT.
- Section 4.3 Table 2, it is hard to tell if the inventions in NI is really beneficial over the ViT baseline, especially because the results are reported from a single run for each setting.

Clarity
- The paper is well-written. Minor comments: notations are a bit confusing. e.g. W_uhij that has four different subscripts merged together.

**Time Spent Reviewing:**

4

---

> ### Author Response · Authors · 2021-08-10
> **Response to Reviewer zsVd**
>
> Thank you for taking the time to review our work. We are glad that you liked the formulation proposed in our work of how to use functions to modulate and recompose computations. In the following we address your concerns and questions in the order they are asked.
>
> **On the rationale behind experiments in Section 4.1**
>
> > Section 4.1 could have baseline models such as the plain transformer model to demonstrate if the inventions in NI is really making the task solvable.
>
> We do not claim that the task is only solvable with our architecture. Indeed, the task can be solved without any modularity, and ViTs can perform reasonably well.
>
> The purpose of the experiment in Section 4.1 is to investigate to what extent new tasks can be solved by recomposing the pre-learned knowledge, given that the original and new tasks share the same primitive operations. We probe this by adapting certain parts of the model to the new task while keeping the others fixed. The appropriate baseline here is therefore the case where all parameters of the neural interpreter are adapted.
>
> Empirically, we find that by adapting just the type inference parameters (i.e. by adapting how the modules are wired), we almost reach the performance of adapting all parameters of the model. Had Neural Interpreters failed to modularize knowledge in a useful way, we would have expected this to be a difficult feat to achieve (which is not the case).
>
> **Continual Learning**
>
> > Additional experiments on continual learning datasets by continuously adding more functions to NI would strengthen the results.
>
> We indeed agree that the design of Neural Interpreters is amenable to continual learning, and identify it as an exciting avenue of future research (line 391). For instance, the property that new functions can be added on the fly harmonizes with existing continual learning methods like orthogonal gradient descent (Farajtabar et al. 2019). However, this warrants a comprehensive exploration which is beyond the scope of this work.
>
> Farajtabar et al. 2019: https://arxiv.org/abs/1910.07104
>
> **Visualizing modularity**
> > Analyzing or visualizing how modular functions are specialized would be beneficial ...
>
> In Figure 4, we visualize the t-SNE embeddings of the inferred types of the set elements as they progress through a Neural Interpreter. We find that the types cluster together in later iterations, and these clusters are picked up by certain functions (depicted by the color of a dot). This suggests that functions indeed learn to specialize, especially when compared against what one would obtain with random routing, as shown in this figure [[https://postimg.cc/14zKbXZ0](https://postimg.cc/14zKbXZ0)].
>
> Nevertheless, this specialization happens in representation space which makes it hard to directly visualize it or even comprehend it on a deeper level.
>
> > It would be also good to analyze how the truncation parameter can be used to control sparsity.
>
> The truncation parameter $\tau$ (Equation. 3) is designed such that it controls the sparsity induced via the modulation mask $C_{ui}$. If $\tau$ is too large, the training starts to be less stable because $C_{ui}$ is quite often sparse (which is desirable from a modularity perspective). On the other hand, if $\tau$ is smaller, then the functions can potentially take more inputs than they need to, but the models converge faster. We encourage the reviewer to look at our answer to reviewer **5mdn**'s question regarding the selection of $\tau$ which sheds more light on its role.
>
> **Adding more layers to ViT**
> >Figure 5 (bottom) could also compare with ViT for example by adding more layers to ViT.
>
> Thank you for the suggestion -- we have carried out this experiment. We fix the first 8 layers of the ViT, add additional layers on top, and finetune these. The results can be seen in this figure [[https://postimg.cc/dDSgxgg1](https://postimg.cc/dDSgxgg1)]. We observe that adding more layers does increase the finetuning performance, as expected. However, Neural Interpreters (augmented with additional functions) outperform ViTs by a considerable margin. We will include this result in the next revision.
>
> **Multiple runs on PGM Tasks**
> > It is hard to tell if the inventions in NI is really beneficial over the ViT baseline, especially because the results are reported from a single run.
>
> Note that training on this data set is rather expensive (each regime taking 6 GPU days). This is likely one of the reasons why prior works also only report a single run [Barrett et al. 2018, Steenbrugge et al. 2018, Wang et al. 2020]. However, we did run the second batch of experiments to add error bars, and the interpretation of the results does not change. The following table presents the mean and standard errors of two independent runs.
>
> | Dataset Regime|VIT (Val)|VIT (Test) |NI(Val)|NI(Test)|
> | :---          |:----:|:---:|:---:|:---:|
> | Triples       |73.05 +- 0.53| 16.28 +- 0.27 |77.76 +- 0.77|19.40 +- 0.42|
> | Attribute Pairs|68.91 +- 0.35|33.86 +-0.17|69.21 +- 0.23|35.61 +- 0.72|
> | Extrapolation |91.33 +- 0.62|17.0+-0.43|92.18 +- 0.23|19.1 +- 0.24|
> | Triple Pairs  |67.81 +- 0.14|44.44 +- 0.24|67.70 +- 0.63|44.70 +- 0.35|
> | Interpolation |88.42 +- 1.04|67.13 +- 0.40|87.75 +- 0.11|69.95 +- 0.39|
> | Neutral       |69.82 +- 2.46|69.2 +- 2.49|74.58 +- 1.92|74.15 +- 2.02|
>
>
> Barrett et al. 2018 https://arxiv.org/abs/1807.04225
>
> Steenbrugge et al. 2018 https://arxiv.org/abs/1811.04784
>
> Wang et al. 2020 https://arxiv.org/abs/2006.11197
>
> **To conclude,** we thank you for your review! If you have additional questions that we can answer, please do not hesitate to let us know in the comments below.

---

> > ### Comment · Reviewer_zsVd · 2021-08-22
> > **Thank you for the response**
> >
> > I thank the authors for their response and additional experiments.
> >
> > I am revising my rating as the response adequately addressed all my concerns.

---

### Official Review · Reviewer_hiUm · 2021-07-16

**Rating:** 7
**Confidence:** 4

**Summary:**

The paper proposes a model based on self-attention, factorised in different modules. It processes a sequence of tokens and could be viewed as a generalisation of a Transformer. The key idea is to have different modules (associated with a function) that behave differently but are general and actually share parameters. The core module of ModAttn can be viewed as a self-attention module with dynamic weights that depend on a context / code c. Using n_f learnable codes, each assigned to a ‘function’ leads to n_f modules, each with different functionality. An important aspect is the fact that each function only affects a subset of tokens, just the ones that are close to a type (learnable parameters) of the function.

**Limitations And Societal Impact:**

This is sufficiently tackled.

**Main Review:**

**Pros**:

+ The paper has an important goal, of obtaining models that are factorized in a meaningful way and offer inductive biases towards reusability and fast adaptation.

+ The method is sound and expands on the works using modular networks and self-attention. The method presents novel elements, the idea of functions that act according to similar dynamics (because they share parameters) but are modulated by the interpreter is novel and interesting.

+ The paper presents some interesting results. Especially interesting is to note that adding multiple functions (Figure 5) improves the results, although they are not pre-trained, and the additional parameters should be small.

+ The model obtains good results on a Abstract Reasoning task that require some form of compositional reasoning.

**Cons**:

- I think that the experimental section is not really convincing enough to strongly support all the claims, as summarised below.

- For Multi-Task Image Classification, the proposed NI model adapts faster than the ViT, but this does not mean that it is modular. The real cause of the improved adaptability could be other than modularity. NI model could simply be easier to optimise, without being modular in any way. And the optimisation could be helped by the fact that NI has 3 times less parameters than ViT.

- For Fuzzy Expressions, it is interesting to note that most of the performance gain is obtained when learning the Type Inference Parameters and this hints that the method learns to recompose the previously learned modules. But this is still a somewhat weaker evidence as the compatibility matrix could encode enough information, that helps the tasks, without necessary reusing components.

- “we include results on a toy problem, where we explicitly probe the ability of Neural Interpreters to learn recomposable computational primitives”. What does ‘explicitly probe’ refer to? Although there are hints that suggest the model leans recomposable primitives, this is not explicitly tested.


- ““there is diversity in how samples are routed through the network, and shows that the routing mechanism discriminates between samples.” Is there a reason why this diversity is different from the diversity obtained at initialisation? Could we interpret this diversity as having more meaning than what would be obtained by taking the similarities of random projections for example?


**Questions**:
Is there a fundamental difference between using multiple LOC or using multiple scripts? Having both functionalities increases the flexibility of the model, making it more capable of expressing various functions more easily, but is there something beyond this?




**Time Spent Reviewing:**

12

---

> ### Author Response · Authors · 2021-08-10
> **Response to Reviewer hiUm**
>
> We are grateful for the effort you have put in to review our work! It is encouraging that you found our method to be "sound" and important parts of it "novel and interesting". In the following we address your comments and questions in the order they were asked.
>
> **Fast Adaptation and Modularity**
>
> > For Multi-Task Image Classification, the proposed NI model adapts faster than the ViT, but this does not mean that it is modular.
>
> Like Bengio et al. 2019, we do not claim that fast-adaptation necessarily implies that the model is modular. Instead, we verify the contraposition: that our architecture (that incorporates modular inductive biases) is indeed capable of fast-adaptation, more so than the non-modular ViT baseline.
>
> > The real cause of the improved adaptability could be other than modularity... the optimization could be helped by the fact that NI has 3 times less parameters than ViT.
>
> Thank you for pointing this out. We have verified that naïvely reducing the number of parameters in VIT (e.g. by reducing width or depth) does not help with fast-adaptation. We reduced the number of parameters in VIT 3 times by (a) reducing its depth from 8 layers to 3 layers (VIT Shallow) and (b) reducing its breadth from 192 dimensions to 68 dimensions (VIT Thin). In both settings, we observed that the adaptation was slower than that of Neural Interpreters [[https://postimg.cc/S2xW0THQ](https://postimg.cc/S2xW0THQ)], especially in the low-sample regime.
>
> This supports the position that for fast adaptation, it is important _how_ parameters are reused, and not just the fact that there are fewer parameters. The former is precisely what the modular inductive biases in NI improve upon.
>
> **Can the compatibility matrix encode enough information to solve the Fuzzy Expression Tasks?**
>
> > The compatibility matrix could encode enough information, that helps the tasks, without necessary reusing components.
>
> The purpose of the compatibility matrix is to route information through the modules, and not to process this information. To verify that this is indeed the case, we repeat the experiment in Table 1, but replace all function codes with that of an arbitrarily selected function. This essentially means that we replace all modules with copies of a single module while training all type inference parameters. If the compatibility matrix alone can encode enough information to solve the task, we would expect this scenario to lead to performance on par with those reported in Table 1 (after tuning type inference parameters).
>
> We find the resulting performance (i.e. coefficient of determination $R^2$) to be around 0.9128 +/- 0.0210, which is significantly worse than the 0.9857 +/- 0.0034 reported in Table 1.
>
> This provides evidence for the hypothesis that the model learns to effectively reuse/rewire the learned modules instead of "merely" encoding information in the compatibility matrix.
>
> **Explicit Probing of Modularity**
>
> > What does ‘explicitly probe’ refer to? Although there are hints that suggest the model leans recomposable primitives, this is not explicitly tested.
>
> By "explicitly probe", we refer to the investigation of what happens when selected parts of the model are kept frozen in Table 1.
>
> We have also performed another new experiment that confirms that the primitives are reused at adaptation time.
>
> In the first phase of this experiment, we pretrain all functions on the digits datasets. Subsequently, we do one of two things: (1) we add and finetune new functions while keeping the previous ones fixed to their pretrained values; (2) we add and finetune new functions while reinitializing the codes and signatures of the previous ones (i.e. by setting them to random vectors).
>
> If the previously trained functions are reused, we would expect setting (1) with pre-trained functions to perform better than (2). This is indeed what we observe, as shown in this figure [[https://postimg.cc/SJ3V6h36](https://postimg.cc/SJ3V6h36)].
>
> **Is there meaningful diversity in how the samples are routed through the network?**
>
> > Is there a reason why this diversity is different from the diversity obtained at initialisation? ...
>
> This is a nice question -- thank you for asking!
>
> To answer it, we visualize the t-SNE embeddings (in Figure 4) for the case where the routing is fixed at initialization. In [[https://postimg.cc/14zKbXZ0](https://postimg.cc/14zKbXZ0)], we see that the type-function assignments (given by the colors assigned to a dot) exhibit less structure and diversity, especially at the later function iterations. This suggests that the learning process indeed induces non-trivial patterns in how information is routed between modules.
>
> **What is the difference between using multiple LOCs vs. multiple Scripts?**
>
> > Is there a fundamental difference between using multiple LOC or using multiple scripts? ...
>
> Multiple LOCs mean that a larger computational unit can be recursively applied (in the function iteration). This could be more appropriate if the primitives are expected to be complex. On the other hand, multiple scripts add depth and is a more versatile way of increasing capacity. When in doubt, we recommend experimenting with the number of scripts, but we also find that the former can work well.
>
> **In conclusion,** we appreciate the effort you have invested towards this review. If there is something you feel we are yet to adequately address, please do not hesitate to engage with us!

---

> > ### Comment · Reviewer_hiUm · 2021-09-02
> > **Post-rebuttal**
> >
> > I thank the authors for their detailed responses to all the reviewers.
> >
> > The rebuttal address some of my concerns and I will increase my score.

---

### Official Review · Reviewer_8zbe · 2021-07-16

**Rating:** 6
**Confidence:** 3

**Summary:**

This paper focuses on improving the generalization ability of deep models to the unseen but related distributions to training data, and thus enhancing interpretability. In specific, a Neural Interpreter is proposed, which factorizes the inference of self-attention networks as the dynamic routing within functions. The Neural Interpreter can flexibly compose computation on a per-sample basis, and it is end-to-end trainable. Some interesting experiments including image classiﬁcation and visual abstract reasoning are conducted to validate the effectiveness of the proposed method.

**Limitations And Societal Impact:**

My concerns on the limitations of this paper have been presented in "Main Review". The authors do not discuss the potential negative societal impact of their work. Adding more related discussions will be helpful.

**Main Review:**

Generally speaking, I think that this is a nice paper. The proposed dynamic Neural Interpreter is novel in concept. The dynamic routing mechanism and the type matching algorithm seem to be natural and flexible. My major concern lies in the experiments.

The comparison with ViT is presented on some tiny data sets like SVHN and MNIST. However, ViT is designed for much larger scenarios like ImageNet. Therefore, I think that current empirical results are not sufficient enough to validate the superiority of Neural Interpreter.

In addition, although ViT recently attracts much attention from the research community, CNN is still widely used in many vision tasks. The proposed Neural Interpreter seems to be a pretty flexible framework. It is possible to be applied to CNN?

**Time Spent Reviewing:**

6 hours

---

> ### Author Response · Authors · 2021-08-10
> **Response to Reviewer 8zbe**
>
> We thank you for taking the time out to review our work. We are glad that you deemed our work to be novel and found the proposed dynamic routing mechanism and the type matching algorithms to be 'natural' and 'flexible'. In the following, we address your questions and concerns in the order they were asked.
>
> >ViT is designed for much larger scenarios like ImageNet. Therefore, I think that current empirical results are not sufficient enough to validate the superiority of Neural Interpreter.
>
> We performed this comparative study while keeping prior works in mind.
> - We train VITs and NIs on the union of three datasets (SVHN, MNIST, and MNISTM) which consists of around 200k samples. In addition, we also make heavy use of data augmentation, following Touvron et al 2020.
> - Cordonnier et al. 2019 show that the VITs can be trained well on a dataset as small as CIFAR10 (similar resolution but ~3x fewer samples compared to our setting).
> -  In our experiments, VITs perform as well as NIs for in-distribution tasks, which is what the models are trained on. This ensures that both models are on equal footing when we compare their speed of adaptation (w.r.t sample complexity) and out-of-distribution generalization.
>
> Touvron et al. 2020: https://arxiv.org/abs/2012.12877
>
> Cordonnier et al. 2019: https://arxiv.org/abs/1911.03584
>
> > The proposed Neural Interpreter seems to be a pretty flexible framework. It is possible to be applied to CNN?
>
> There are many promising ways by which Neural Interpreters can be made to work with CNNs. One such way is to use features extracted by a CNN instead of the raw patch embeddings (as done in ViTs). Another interesting approach could be the one considered in DeformableDETR (Zhu et al. 2020), where a convolutional network is used as a multi-scale backbone.
>
> Zhu at al. 2020 https://arxiv.org/abs/2010.04159
>
> > The authors do not discuss the potential negative societal impact of their work ...
>
> Thank you for your suggestion! We will expand on our discussion about potential negative societal impacts in the next revision.
>
> **In conclusion**, thank you again for reviewing our work! Should you have additional questions, please do not hesitate to engage with us.

---

> > ### Comment · Reviewer_8zbe · 2021-09-12
> > **Thank you for the response.**
> >
> > Thank you for the detailed response. The rebuttal address my concerns and I decide to keep my score (weak acceptance).

---

### Official Review · Reviewer_5mdn · 2021-07-16

**Rating:** 8
**Confidence:** 4

**Summary:**

This paper proposes a method for modularizing neural networks inspired by programming functions and compilation. Building on the success of attention and transformer architectures, the neural interpreter (NI) proposes a new architecture built on layered functions, each of which is composed of stacked MLP and attention layers. Functions and input variables both have embeddings, called “signature and type” respectively, that allow NIs to learn specialized functions that operate over a subset of inputs. The NI is applied to a suite of image datasets as well as to Raven matrices, where it is shown to outperform vision transformers.

**Limitations And Societal Impact:**

I think an explicit limitations section would be helpful to (1) be transparent about any optimization or modeling difficulties in NI and also (2) explore cases in which NI does not perform well. The broader impact statement could be expanded to highlight the importance of modularity of neural networks for maintaining and improvement models.

**Main Review:**

Originality:
The idea is very creative and presents a scalable, continuous approximation to modular components in a neural network. In particular, I found the joint mapping of functions and inputs in an embedding space to an insightful idea that theoretically enables the neural network to compartmentalize information and reuse computation. Overall, I enjoyed the analogy to code interpreters and found the presentation compelling. My primary concern from a modeling perspective is that it was not clear to me why functions are encouraged to specialize to specific inputs e.g. there is not objective encouraging disentanglement or diversity, so why should the functions not all learn similar, or overlapping things -- which would limit the modularity and reusability of the network.

Quality/Significance:
The experiments did a good job demonstrating promising results for the NI, although improvements can be made. For instance, in the toy experiment in 4.1, it would be nice to see adaptability to held-out base operations (e.g. XOR). In 4.2, I would like to also compare NI and VTs through more naturalistic images. Consider exploring the Meta-dataset as a collection of realistic images. I would have also liked to see more evidence in Figure 5 on functions learned in pretraining being reused, or more analysis in section 4.3 on why NI is outperformed by WReN in the Attribute P. and Triples datasets.

Clarity:
Very clear writing. The authors do a good job of describing a complex system clearly. I appreciate explicitly denoting changes in notation (e.g. line 123-125), and sentences in Section 2 explaining roles.

Questions about Section 2:
- Line 73: are the image embeddings learned as well? Or pretrained?
In functions, it seems like you want to disentangle information between the code vector and the type vector. How do you ensure the type vector captures no information about the code and vice versa?
- In type matching, you want to have function signatures and element types “cluster” naturally such that some functions specialize to some types, etc. To my understanding though, it seems like nothing prevents all the type vectors and function signatures from being close to each other such that all functions have access to all types. Where is diversity / sparsity being enforced? A related question: how do you prevent two functions from learning the same thing?
- How do you choose tau? If it is too high, I imagine learning will be very slow as C_{u,i} will be 0 for most matches. But at the same time, you want tau to be big in order to specialize functions (i.e., so that functions don’t take everything as input).
- Do you enforce type embeddings and signature embeddings to live in a compact space (e.g. surface of a hypersphere?), otherwise I imagine you could have issues with unbounded vector magnitudes in training.
ModLin and ModMLP remind me a lot of FiLM (Perez et. al.). Might be worth citing or comparing against.
- Conditioning attention networks (transformers) on vectors has been studied in the past: popular methods include an adapter network (Houlsby et. al.), FiLM, and a simple task token (TAM, Logeswaran et. al.). I’m curious how ModAttn compares to these.
- When adding a new function (line 158), do you need to finetune old functions or can you keep them fixed? I could see this either way since existing functions might “take up most of the area in the embedding space”, it becomes harder to add new functions without moving around old functions to make space.
- I might have missed this but are the conditioning vectors c_u learned? If so, how? Through a network? What are the inputs it gets?
- Is it possible to come up with a synthetic example of something the neural interpreter can easily capture with its architecture that stacked transformer layers cannot? This would help me clarify when the interpreter should be superior, and why.

Questions on Section 3:
- Consider adding a citation to Neural Event Semantics (Buch et. al.) in the related work.

Questions on Section 4:
- Building on the toy experiment in Section 4.1, can the neural interpreter adapt to a held out operation? (e.g. it does not get to pretrain on the xor operation and must quickly learn new functions post training). This would be more convincing to show generalizability of modular components.
- In the multi-task vision experiments, is fine-tuning changing just the type inference parameters or all parameters? Iit would be nice to see that most of the learned functions from pretraining don’t change, hence “reusing modularized knowledge”. Do you know how much information about previous tasks is preserved after adaptation? One could test the performance of the NI on pretrianing tasks after adapting to K-MNIST to see how much information is forgotten. I’m also interested in comparing the orange line in Figure 5 to a NI trained from scratch on K-MNIST (how fast does that learn?).
- In Table 2, we see in some tasks, WReN / VAE-WReN outperform ViT and NI: do you have a sense of why for Attribute P. and triples, this is true? Is there something fundamental to those tasks that requires a level of disentanglement that NI does not have? Similarly, I imagine NI would benefit greatly from function disentanglement.


**Time Spent Reviewing:**

7

---

> ### Author Response · Authors · 2021-08-10
> **Response to Reviewer 5mdn (part 1)**
>
> Thank you for the time you have invested for a detailed and constructive feedback! We are elated that you find our idea _very creative_, components of it _very insightful_ and the presentation _compelling_. Your many questions and comments have led us to some valuable insights (detailed below) which strengthen our work. In the following we address these in the order they were asked.
>
> **Why should functions learn to specialize or be diverse without any specific objective enforcing it?**
>
> > Why functions are encouraged to specialize to specific inputs. Why should the functions not all learn similar, or overlapping things.
> >...
> >Where is diversity / sparsity being enforced? A related question: how do you prevent two functions from learning the same thing?
>
> Thanks for asking -- this is an interesting detail that will be more prominent in the next revision. We experiment with two ways of enforcing diversity and sparsity.
>
> The most effective one is to fix the signatures at initialization, and have the type inference MLP adapt to them (during pre-training). Note that this does not overly constrain the model, since the learned type inference MLP should be able to map two entities that should be processed by the same function to type vectors close to the (fixed) signature of the said function. A discussion can be found in Appendix A.1.4, line 637 et seq.
>
> In our initial experiments, we also experiment with a hinge-loss term that repels types or signatures from each other. This is not as effective, and in some cases, it prevents the formation of clusters in type-space (like the ones shown in Figure 4 right). We, therefore, avoid using it in later experiments.
>
> **Adaptability to held-out base operations (e.g. XOR).**
>
> >It would be nice to see adaptability to held-out base operations (e.g. XOR).
>
> Thank you for the suggestion. We have run this experiment: following the experimental protocol described in Section 4.1, we first pretrained the model on expressions featuring the OR primitive and then finetuned it on the held-out expressions featuring the XOR primitive.
>
> The results are as follows (comparable with Table 1 in paper):
>
> - $R^2$ after finetuning CLS Tokens: `0.9171 +/- 0.0196`
> - $R^2$ after finetuning Type Inference: `0.9835 +/- 0.0030`
> - $R^2$ after Finetuning All: `0.9953 +/- 0.0021`
>
> This hints that the model has learned primitives that facilitate expressing the XOR operation, which is not entirely surprising given that XOR itself can be expressed with OR, AND, and NOT operations. We will include this experiment in the next revision.
>
> **Experiments on more naturalistic datasets.**
>
> >In 4.2, I would like to also compare NI and VTs through more naturalistic images.
>
> On the one hand, we agree that scaling our experiments to bigger and more naturalistic images would strengthen our paper. But on the other hand, this presents its own set of engineering / compute challenges and design choices. A thorough investigation (not only on meta-dataset, but also VTAB and WiLDS) atop existing results would breach the scope of this work and is orthogonal to the focus of the experiments presented in this paper.
>
> Following the request and to verify that the method shows promise for more natural images, we did carry out some preliminary experiments on FMOW-WiLDS, a dataset of satellite images. We chose FMOW-WiLDS because (a) it comes with a realistic OOD test dataset (b) the images therein may contain motifs (e.g. buildings, roads) that appear in test and train distributions in different combinations. Due to (b), we would expect a modular model like NI to outperform the non-modular baseline (ViT) out-of-distribution (cf. lines 298-300 in the manuscript).
>
> Indeed, we find that NI outperforms the ViT by 1.6% out-of-distribution. We verify that in-distribution, the ViT performs on par with NI (ViT is better by 0.5%), implying that there are no optimization issues confounding the results. Note, however, that due to time and compute constraints for the rebuttal, we had to use (for both models) a large patch-size (28 x 28 instead of 16 x 16) to reduce the number of tokens, given the high resolution of the input images. This is a serious caveat, and consequently, we cannot match the state of the art.
>
> In light of the discussion above, we are inclined to not include these experiments in the paper, unless you strongly recommend us to.
>
> **More Evidence of Function Reuse**
>
> >I would have also liked to see more evidence in Figure 5 on functions learned in pretraining being reused.
>
> We have devised an experiment to gather more evidence of function reuse.
>
> In the first phase of this experiment, we pretrain all functions on the digits datasets. Subsequently, we do one of two things: (1) we add and finetune new functions while keeping the previous ones fixed to their pretrained values; (2) we add and finetune new functions while reinitializing the codes and signatures of the previous ones (i.e. by setting them to random vectors).
>
> If the previously trained functions are reused, we would expect setting (1) with pre-trained functions to perform better than (2). This is indeed what we observe, as shown in this figure[[https://postimg.cc/SJ3V6h36](https://postimg.cc/SJ3V6h36)]. We will include this experiment in the next revision.
>
> > Line 73: are the image embeddings learned as well? Or pretrained?
>
> Yes, we learn the embeddings. However in our ablations, we also find that, like in prior works (e.g. Chen et al. 2021), random embeddings also work well.
>
> Chen et al. 2021: https://arxiv.org/abs/2104.02057
>
> **On the Role of Signature / Type and Code Vectors**
>
> >How do you ensure the type vector captures no information about the code and vice versa?
>
> The signature and type vectors only influence computation via the type matching mechanism, i.e. in equation 3. They determine _where_ information is routed, i.e. which modules receive what information.
>
> The code vectors, on the other hand, influence computations by conditioning linear layers (as described in equation 4). In doing so, they determine _how_ information is processed inside the modules that receive it. In particular, the code vector conditions all operations inside a function (all layers in all LOCs, for a given function).
>
> In summary, the signature / type vectors and code vectors have entirely different roles. Nevertheless, it is necessary that for a given function, the signature vector is appropriate for the respective code vector.
>
> **Choosing Tau**
>
> >How do you choose tau? ...
>
> Your intuition is indeed correct -- on the one hand, if $\tau$ is too large, the training starts to be less stable because $C_{ui}$ is quite often sparse (which is desirable from a modularity perspective). On the other hand, if $\tau$ is smaller, then the functions can potentially take more inputs than they need to, but the models converge faster.
>
> Our heuristic, therefore, is to pick the largest $\tau$ that is fast to learn, given the compute budget. The value of this $\tau$ can depend on the data domain, but it can easily be found with hyper-parameter optimization. We also provide ablation for selecting useful $\tau$ on multi-task image classification experiment in Figure 9.
>
> **Where do type and signature embeddings live?**
>
> > Do you enforce type embeddings and signature embeddings to live in a compact space (e.g. surface of a hypersphere?) ...
>
> Yes -- the type and signature vectors live in a hypersphere (i.e. they have unit norm), precisely due to the reason you mention. This is mentioned in line 98.
>
> **Precise differences to some prior works**
> > Conditioning attention networks (transformers) on vectors has been studied in the past ... I’m curious how ModAttn compares to these.
>
> Thank you for pointing us to these works, we will include them as related work in the next revision. In the following, we expand on the similarities and differences between these methods.
>
> 1. _Adapter Network (Houlsby et al.)_. Like us, Adapter Networks add additional parameters that can be trained at adaptation time for the downstream task. However, unlike us, the new parameters used at the adaptation time affect all tokens. In ModAttn, the new modules / functions can select which tokens they want to operate on. ModAttn therefore explicitly incorporates the inductive bias of sparsity via the kernel modulation.
>
> 2. _FiLM (Perez et al.)._ The ModLin layer that we adapt from [CIPS](https://arxiv.org/abs/2011.13775) is indeed somewhat similar to a FiLM layer. The differences are the following:
>     * In the language of FiLM, we only multiply the inputs with gamma (predicted weights), and omit the addition of a predicted bias term.
>     * Unlike in the FiLM paper, the gammas in this work are not obtained from the input itself, but a code vector that is freely learned (but shared between layers).
>
> 3. _TAM (Logeswaran et al.)._ The multi-task learning aspect of our work (i.e. using multiple CLS tokens, one per task) indeed shares the task-conditioning mechanism with TAM (Logeswaran et al.), and we will be sure to attribute it as such. Nevertheless, the core contribution of our work is modularity, which is complementary to the problem addressed by Logeswaran et al.
>
> > Consider adding a citation to Neural Event Semantics (Buch et. al.) in the related work.
>
> Thanks for the suggestion. We will make sure to include it in the next revision.
>
> **Are old functions fixed when fine-tuning?**
>
> > When adding a new function (line 158), do you need to finetune old functions or can you keep them fixed? ...
>
> Our intuition aligns with yours. We run additional experiments to verify that both options can work [[https://postimg.cc/QVVL8X3s](https://postimg.cc/QVVL8X3s)] (the numbers at 10 epochs are comparable with those in Figure 5 bottom, where we finetune all functions). Consistent with expectations, we find that finetuning only new functions incurs some loss in performance, e.g. ~55% vs. ~69% after finetuning for 10 epochs on 8192 samples of K-MNIST.

---

> > ### Author Response · Authors · 2021-08-10
> > **Response to Reviewer 5mdn (part 2)**
> >
> > **Are the Conditioning Vectors Learned?**
> >
> > > I might have missed this but are the conditioning vectors c_u learned? ...
> >
> > Yes, the conditioning / code vectors $c_u$ are learned as free-parameters.
> >
> > **What can Neural Interpreters do that ViTs cannot?**
> > > Is it possible to come up with a synthetic example of something the neural interpreter can easily capture with its architecture that stacked transformer layers cannot? ...
> >
> > Thank you for pushing us to think about this!
> >
> > **Dynamic Computational Cost after Training.** We have found that with Neural Interpreters, we can dynamically decrease the amount of compute _after_ training, e.g. at inference time. This can be useful in settings where a pre-trained Neural Interpreter is deployed to various computational environments, and there is no equivalent procedure that we're aware of for a conventional transformer.
> >
> > In this figure [[https://postimg.cc/68xn65Fm](https://postimg.cc/68xn65Fm)], we show that the recurrence over function iterations can be reduced at test-time (thereby reducing the inference speed) while only gradually degrading performance. In some scenarios, we find that at even half the amount of compute the model was trained with, the original performance (on test set) is preserved.
> >
> > We make a similar observation while removing modules (thereby reducing inference speed and the memory cost): in this figure [[https://postimg.cc/9ztjgxCw](https://postimg.cc/9ztjgxCw)], we show how removing modules (also after training) also only gracefully degrades the performance, further supporting the hypothesis that Neural Interpreters are modular.
> >
> > These experiments will be included in the next revision of the main paper.
> >
> > **What changes during finetuning in the multi-task vision experiments? Is knowledge being reused?**
> >
> > > In the multi-task vision experiments, is fine-tuning changing just the type inference parameters or all parameters? ...
> >
> > In the multi-task experiments (Section 4.2), we finetune all the parameters. This is to be consistent with VIT, where we have no choice but to finetune all parameters.
> >
> > _Is modularized knowledge being reused?_ As mentioned above (under "More evidence of Function Reuse"), we do provide additional evidence that modularized knowledge is indeed being reused.
> >
> > **How much information about the pre-training tasks is forgotten after fine-tuning?**
> >
> > > Do you know how much information about previous tasks is preserved after adaptation? ...
> >
> > We have carried out additional experiments to answer these questions. The setup is as follows: we fix the pretrained functions, add new functions and adapt these for a various number of epochs (with 8192 samples). We track the performance on the pre-training (digits) tasks after adaptation to K-MNIST.
> >
> > The results are summarized in this figure [[https://postimg.cc/QVVL8X3s](https://postimg.cc/QVVL8X3s)]. We find that some -- but not all -- information about the pretraining tasks (MNIST, MNISTM and SVHN) is lost as the model adapts to K-MNIST. In particular, we find that MNIST suffers more than MNISTM and SVHN, which can be explained by the fact that K-MNIST and MNIST are visually similar: the model might repurpose some "circuits" developed for MNIST in order to adapt to K-MNIST, while disturbing the circuits for SVHN and MNISTM to a lesser extent.
> >
> > The latter is encouraging from the perspective of continual learning, but there are additional design decisions to be made in order to fully exploit the potential of Neural Interpreters. Future work will explore synergies with Orthogonal Gradient Descent, which we intuit to harmonize well with the inductive biases of Neural Interpreters. We will include this discussion in the next revision.
> >
> > **How fast NIs adapt while training from scratch?**
> > >I’m also interested in comparing the orange line in Figure 5 to a NI trained from scratch on K-MNIST (how fast does that learn?)
> >
> > We investigate this here [[https://postimg.cc/tY375dmB](https://postimg.cc/tY375dmB)]. NIs are slow to adapt when trained from scratch.
> >
> > **Why does the VAE-WReN outperform ViT and NI?**
> > >In Table 2, we see in some tasks, WReN / VAE-WReN outperform ViT and NI: do you have a sense of why for Attribute P. and triples, this is true? ...
> >
> > While ViT and NI both outperform WReN, no method that we are aware of has been able to outperform the numbers reported by Steenbrugge et al. for VAE-WReN on Attribute Pairs and Triples. We find this surprising and do not have an explanation, but our hypothesis is that the inductive bias of feature disentanglement (via VAE) plays an important role. This does not, however, explain why it does not help as much for the other tasks.
> >
> > **To conclude,** we thank you again for the effort you have invested in this review. If there is something you feel we have not adequately addressed yet, please do not hesitate to let us know!

---

> > > ### Comment · Reviewer_5mdn · 2021-08-16
> > > **Thank you for the detailed response!**
> > >
> > > Thanks to the authors for responding to my questions in great detail and providing new experiments. I enjoyed reading them very much! A few more thoughts in response:
> > >
> > > - The XOR experiments are quite nice (i hope they make it into the appendix at the least). Generalization to XOR from OR, AND, and NOT components is not trivial and in my opinion, adds to the experiment.
> > >
> > > - The experiment showing pretrained functions outperform re-initialized functions is also interesting! Thanks to the authors for running this.
> > >
> > > - I agree that using a naturalistic image experiment to show scaling is  somewhat orthogonal to the method being proposed, but at the same time would find the results a great proof of the method's capability.
> > >
> > > - I don't have strong opinions on whether to include the FMOW-WiLDS dataset and leave it up to the authors' discretion (as I do not know how much of an impact using a different patch size makes).
> > >
> > > - Do you find that the appropriate $\tau$ varies by dataset? Some heuristical guidance on how to choose this $\tau$ would be helpful.
> > >
> > > - The dynamic computational cost point is interesting -- i would consider pointing out this distinction to the ViT in the main paper.
> > >
> > > - Interesting that inductive bias of feature disentanglement could be the reason VAE-WREN does so well. Can you apply similar disentanglements to NI (via function embeddings)?
> > >
> > > Overall, I quite like this paper and find its contribution significant.

---

> > > > ### Author Response · Authors · 2021-08-18
> > > > **Thank you for your response!**
> > > >
> > > > Thank you for your response! We are glad that you enjoyed reading our response and about the additional experiments we conducted based on your feedback.
> > > >
> > > > > The XOR experiments are quite nice (i hope they make it into the appendix at the least). Generalization to XOR from OR, AND, and NOT components is not trivial and in my opinion, adds to the experiment.
> > > >
> > > > We will indeed include the XOR results in the next revision!
> > > >
> > > > > Do you find that the appropriate $\tau$ varies by dataset? Some heuristical guidance on how to choose this $\tau$ would be helpful.
> > > >
> > > > We find that an appropriate $\tau$ can vary with the dataset, but only to a lesser extent. For instance, in datasets with less background clutter (e.g. MNIST), we find that smaller values of $\tau$ can also consistently lead to good in-distribution (validation) performance (cf. Figure 9 in Appendix C.3).
> > > >
> > > > For low compute budget hyper-parameter sweeps, we recommend sweeping $\tau$ between 1.4 and 1.6. In our experiments, we find that values of $\tau$ in this range tend to perform well for all datasets we experiment on. For larger compute budgets, we recommend sweeping between 1.2 and 1.7, as recommended in line 634 of Appendix A.1.3.
> > > >
> > > > > Can you apply similar disentanglements to NI (via function embeddings)?
> > > >
> > > > For PGM tasks, it is indeed possible to feed the NI with disentangled features of input images (cf. lines 368 and 369). The resulting model (VAE-NI) can then be fairly compared with VAE-WReN, and we anticipate some improvements analogous to those obtained by VAE-WReN over WReN. However, we defer exploring this direction to future work because it has to do with the feature extraction part of the pipeline, which is entirely complementary to the main strength of NIs that we wish to demonstrate with the PGM experiments, namely, compositional reasoning.
> > > >
> > > > > Overall, I quite like this paper and find its contribution significant.
> > > >
> > > > We are glad that you do, and we are grateful for your review! Your comments are proving quite valuable as we build the next revision, and they have encouraged us to explore directions that we would have otherwise missed.

---

### Author Response · Authors · 2021-08-11
**General comment to all reviewers**

We are grateful to all reviewers for the generous effort they have invested in reviewing our work, and are enthused to learn that they find our ideas “very creative”, “insightful”  (Reviewer 5mdn), “novel in concept”, “natural and flexible” (Reviewer 8zbe), “sound” with “novel elements” (Reviewer hiUm) and “promising” (Reviewer zsVd).

We appreciate the valuable feedback, and take this opportunity to highlight a few important enhancements we have made during the rebuttal period. Details and additional experiments can be found under individual comments to the reviewers.

* Reviewer 5mdn encourages us to think about what makes the Neural Interpreter (NI) architecture special in comparison to a stack of transformer layers. We respond with experiments showing that NIs support dynamical computational effort **after** training. In other words, at inference time and without any additional training, one may choose to run fewer (recurrent) iterations of the model or to drop modules, and have the performance degrade gracefully. We imagine this to be a useful feature for deployment to diverse devices with varying compute capabilities. It also demonstrates the potential of modular inductive biases central to the design of NIs.

* Reviewers 5mdn and hiUm request more evidence to demonstrate that the modular primitives learned by NI are indeed reusable. We respond with additional experiments showing that at adaptation time, if one does not use pre-trained primitives (and instead use random ones), the performance degrades. This demonstrates that the primitives learned during pre-training are indeed useful when adapting to a new task.

* A core component of the NI architecture is the dynamic and differentiable routing of information through the network. Reviewer hiUm asks whether such a learned routing mechanism is more meaningful than a random one. We respond with a figure comparing the t-SNE visualizations of type space (cf. Figure 4 in the manuscript) when the routing is learned vs. when it is random. We find that in the former, a large number of clusters emerge in type space (especially at later iterations). Further, elements in a cluster tend to be most strongly matched to the same function, and all functions are utilised. Such structure is missing in the latter (i.e. when the routing is random), where we find that a few functions capture most inputs (i.e. the functions are not adequately utilised) and there are not many clusters in type space. This demonstrates that the learning process indeed induces interesting patterns in how information is routed through the network.

* To obtain Figure 5 (bottom), we adapted the codes and signatures of all functions to K-MNIST. Reviewer 5mdn asked whether we can keep the old functions fixed to their pre-trained state and adapt only new functions to the new task. We show that it is indeed possible, while incurring a reasonable loss in performance (which is expected, since the model is afforded less flexibility).

* Reviewer 5mdn asks whether information about pre-training tasks (e.g. MNIST, SVHN and MNISTM) is retained after adaptation to a new task (e.g. K-MNIST). We run this experiment to find that some information is indeed preserved, especially about the tasks that are dissimilar to the new adaptation task. One possible explanation for this phenomenon is that the “routing circuits” developed for the pre-training task similar to the new task (here, MNIST) is repurposed when the model is adapted to the new task (here, K-MNIST), while disturbing such circuits for other, less similar tasks (SVHN and MNISTM) to a lesser extent. We believe this is encouraging for future work on continual learning using Neural Interpreters, especially in conjunction with methods like Orthogonal Gradient Descent.

---

### Decision · Program_Chairs · 2021-09-27

**Decision:**

Accept (Poster)

**Comment:**

This paper addresses the goal of modular information processing in deep networks in an interesting way, inspired by the architecture of function application in a PL interpreter. Everyone agrees that this is an interesting approach, and the analogy to interpreters is useful. The results, especially after the response phase, support the ability of this architecture to generalize in interesting ways. It would be useful in revision to make sure to clarify some key points raised by reviewers (such as what encourages the NI to learn sparse types).